# Coupled symmetric and asymmetric circuits underlying spatial orientation in fruit flies

Ta-Shun Su[1], Wan-Ju Lee[1], Yu-Chi Huang[1,2], Cheng-Te Wang[1,2] & Chung-Chuan Lo [1,2,3]

Maintaining spatial orientation when carrying out goal-directed movements requires an animal to perform angular path integration. Such functionality has been recently demonstrated in the ellipsoid body (EB) of fruit flies, though the precise circuitry and underlying mechanisms remain unclear. We analyze recently published cellular-level connectomic data and identify the unique characteristics of the EB circuitry, which features coupled symmetric and asymmetric rings. By constructing a spiking neural circuit model based on the connectome, we reveal that the symmetric ring initiates a feedback circuit that sustains persistent neural activity to encode information regarding spatial orientation, while the asymmetric rings are capable of integrating the angular path when the body rotates in the dark. The present model reproduces several key features of EB activity and makes experimentally testable predictions, providing new insight into how spatial orientation is maintained and tracked at the cellular level.

[1] Institute of Systems Neuroscience, National Tsing Hua University, Hsinchu 30013, Taiwan. [2] Institute of Bioinformatics and Structural Biology, National Tsing Hua University, Hsinchu 30013, Taiwan. [3] Brain Research Center, National Tsing Hua University, Hsinchu 30013, Taiwan. Correspondence and requests for materials should be addressed to C.-C.L. (email: cclo@mx.nthu.edu.tw)

Navigation in the environment requires that an animal keeps track of its spatial orientation. Furthermore, when the animal loses visual contact with external cues, the animal needs to retain a short-term memory of its orientation and continuously updates this representation during the movement. Recent studies in *Drosophila melanogaster* have demonstrated that the ellipsoid body (EB) of the central complex[1–8] exhibits localized activity (an activity bump) that represents the direction of the most salient visual cue[9, 10] (Fig. 1a). This activity bump persists even in darkness and produces counter-movements in response to horizontal rotation of the body (Fig. 1b). This "bump-shifting" function suggests that the activity bump represents an "internal compass", which enables the fly to keep track of its spatial orientation[10, 11].

The persistent activity bump observed in the EB is surprisingly consistent with that predicted in attractor neural circuit models of spatial working memory and decision-making[12–14]. In these models, a working memory of a spatial location is associated with localized neuronal activity within a network characterized by local recurrent excitation and global feedback inhibition (Fig. 1c). In decision-making models, only one activity bump corresponding to the most salient cue will develop due to the winner-take-all dynamics of the attractor neural network[14].

However, a study by Seelig and Jayaraman[10] indicated that the functional features of the EB cannot be fully explained by models of simple working memory or decision-making networks. Seelig and Jayaraman[10] observed that the bump continued to update its location in accordance with horizontal rotation of the body in darkness, suggesting that the EB and its associated brain regions perform neural computation of angular path integration based on self-motion cues. Such observations suggest that EB neurons exhibit functional properties similar to those observed in the head-direction cells of rodents[15–23]. Although several theoretical studies for rodents[19, 24–29] and insects[30, 31] have provided valuable insights into how the head-direction system or motion integration may work, these studies are based on hypothetical network structures. Moreover, the single-cell-level connectomes of related brain regions in rodents and in most insects (except for *Drosophila*) are not available. Therefore, it remains unclear which (if any) network model accurately describes the actual circuit mechanism underlying angular path integration.

To address this issue, we examined recently published anatomical data and theoretical analysis regarding the *Drosophila* central complex[32–34] and constructed the connectome for neurons connecting the EB and protocerebral bridge (PB). The results of our analysis indicated that, from a network architecture perspective, the EB–PB circuit is capable of maintaining an activity bump and performing angular pathway integration, further suggesting its role in the maintenance of spatial orientation. We constructed a computational model of the EB–PB circuitry and demonstrated that the simulated neural activity reflected several key features of empirically observed neural dynamics[10]. We also discuss the specific predictions associated with the proposed model.

## Results

**EB–PB circuitry**. We constructed a model of EB–PB circuitry by combining cellular-level anatomical data from two recently published papers[32, 33] (Supplementary Table 1). Three classes of neurons—EIP, PEI, and PEN—constitute the complex circuits

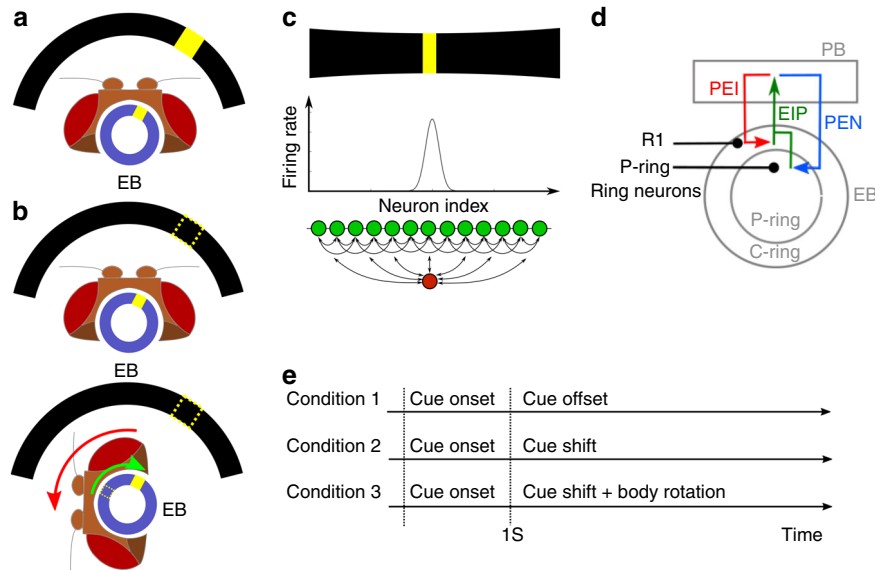

**Fig. 1** The maintenance of spatial orientation and the ellipsoid body (EB) circuit in *Drosophila*. **a** A schematic of a fly head and EB (*blue circle*), which can be mapped to 360° of the visual field (partially represented by the *black arcs*). When a visual cue (the *yellow bar* in the visual field) is presented, the EB responds to it with localized activity, or the activity bump (*yellow square* in EB). **b** *Top*: When a visual cue is removed (*dashed bar*), the activity bump persists, representing the memory regarding the orientation of the visual cue. *Bottom*: When the fly changes its orientation in darkness, the activity bump shifts accordingly, indicating the ability to maintain spatial orientation in the dark[10]. **c** An activity bump, which is localized neural activity (*middle*) in response to a visual stimulus (*top*), is considered as a neural representation of working memory of the task-relevant stimulus if the bump persists after the offset of the stimulus. Theories suggest that the activity bump can be maintained by a network with local recurrent excitation and global feedback inhibition (*bottom*). *Green circles*: excitatory neurons. *Red circle*: inhibitor neurons. *Arrows*: synaptic connections. **d** The circuit considered in the study consists of three classes of observed neurons (PEI, PEN, and EIP) that connect the EB and protocerebral bridge (PB). EIP neurons project from the EB (both C and P rings) to the PB and form two feedback loops with the PEI and PEN neurons separately. We also consider ring neurons (*black*) that innervate the C-ring (R1) or P-ring. **e** We performed the simulated spatial orientation task under three conditions. For all three conditions, the cue was turned on at $t = 0$ s and remained for 1 s. In the first condition, the cue was turned off at $t = 1$ s. In the second condition, the cue remained on but its location (azimuthal angle) began to shift after $t = 1$ s. In the third condition, the cue was turned off at $t = 1$ s, after which the fruit fly body began to rotate horizontally

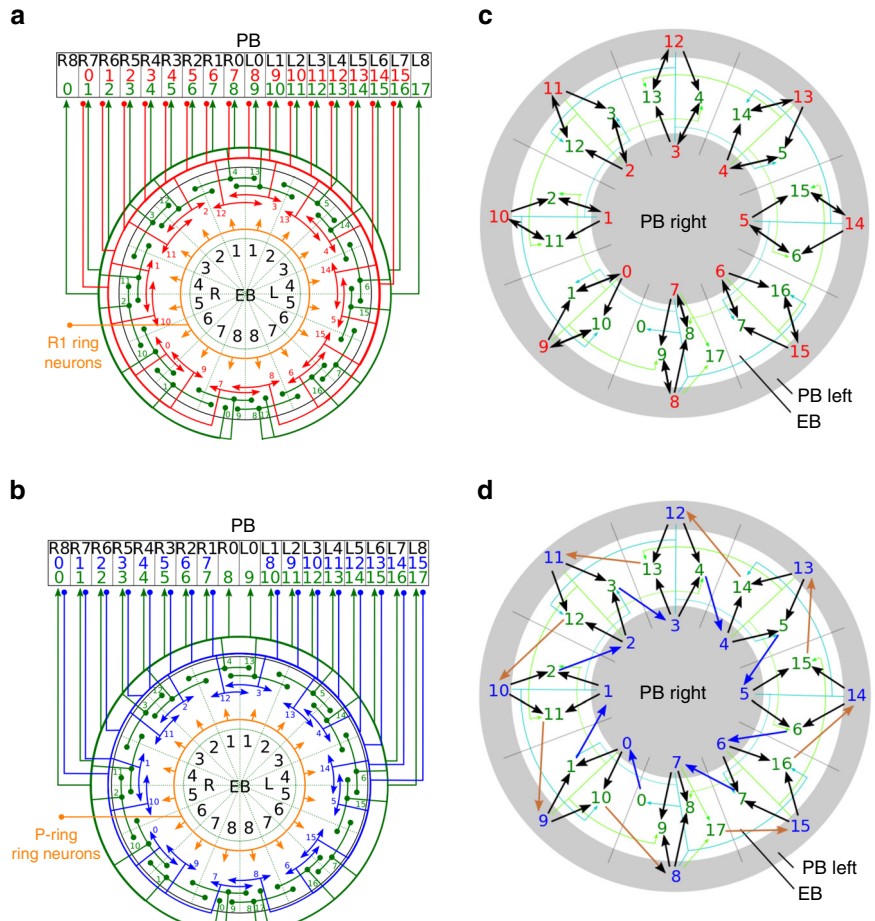

**Fig. 2** The circuit structures of the C and P rings. **a** The C-ring and **b** the P-ring circuits represented by the neuronal innervation reconstructed based on data from recent anatomical studies[32, 33]. The EIP, PEI, PEN, and ring-neuron classes are represented by *green*, *red*, *blue* and *orange lines*, respectively. *Arrowheads* indicate the axonal terminals, while *solid circles* indicate the dendritic terminals. The *numbers* denote the neuron types for each color-matched neuron class. The *black numbers* label the regions in each neuropil. **c** The C-ring and **d** the P-ring circuits in a representation depicting both neuron connections and innervation regions. To visually emphasize the excitatory feedback circuits, the ring neurons are not included. The outer and inner circles represent the left and right sides of PB, respectively, while the middle ring is for the EB. Each neuron (as indicated by its color and type number, same as in **a** and **b**) is placed in its main dendritic innervation region. Arrows indicate the connections between neurons, with the thickness qualitatively representing the synaptic strength. The arrows are colored for the visualization purpose. The C-ring circuit is mainly constructed by symmetric feedback connections, while the P-ring circuit is characterized by asymmetric connections, which can be separated into a clockwise (*blue arrows*) and a counterclockwise (*brown arrows*) subcircuit

that connect the EB and PB. Neurons in the EIP class project from the C and P rings (two subdivisions of EB, see Methods) of the EB to the inferior dorsofrontal protocerebrum (IDFP) and PB. PEI neurons project from the PB back to the EB (C-ring) and IDFP. PEN neurons project from the PB back to the EB (P-ring) as well as to the noduli (NO). Therefore, neurons of the EIP class form two coupled feedback circuits, one with the PEI neurons in the PB and EB C-ring, and the other with the PEN neurons in the PB and EB P-ring (Fig. 1d). The model also includes two types (R1 and P rings) of ring neurons that project from the lateral triangle to all regions in the C and P rings, respectively[35, 36]. We hypothesized that the ring neurons can be classified into three functional types and each separately makes contacts with EIP, PEI, and PEN neurons. The model was tested for its capability of spatial orientation in three task conditions (Fig. 1e) (see Methods).

**A novel representation of the EB–PB circuits**. Based on the anatomical data[32, 33], each of the three neuron classes can be further divided into several neuron types, which innervate

different regions of the EB and PB and form complex circuits (Figs. 2a, b). The structures of EIP–PEI and EIP–PEN circuits differ in the following aspects: first, the PEN and PEI neurons that originate from the same PB region project to different regions in the EB; and second, PEI neurons innervate PB regions 0–7 on both sides (R0–R7 and L0–L7), while PEN neurons innervate PB regions 1–8 on both sides (R1–R8 and L1–L8) (Figs. 2a,b). However, it is difficult to infer the functional differences between the two circuits only through inspection of the representation in Figs. 2a,b, which provides valuable information regarding the regional innervation, yet offers little in terms of connectivity among these neurons. To solve this issue, we replotted the circuit in a novel way to convey information regarding both innervation and connectivity. We presented the circuit in three concentric circles, which, from the inside toward outside, represent PB right side (R0–R8), EB, and PB left side (L0–L8), respectively. We placed each neuron type in the regions where its dendrites are located and indicated the axonal projections by arrows (Figs. 2c, d). The plot revealed several intriguing features of the C-ring and P-ring circuits as follows.

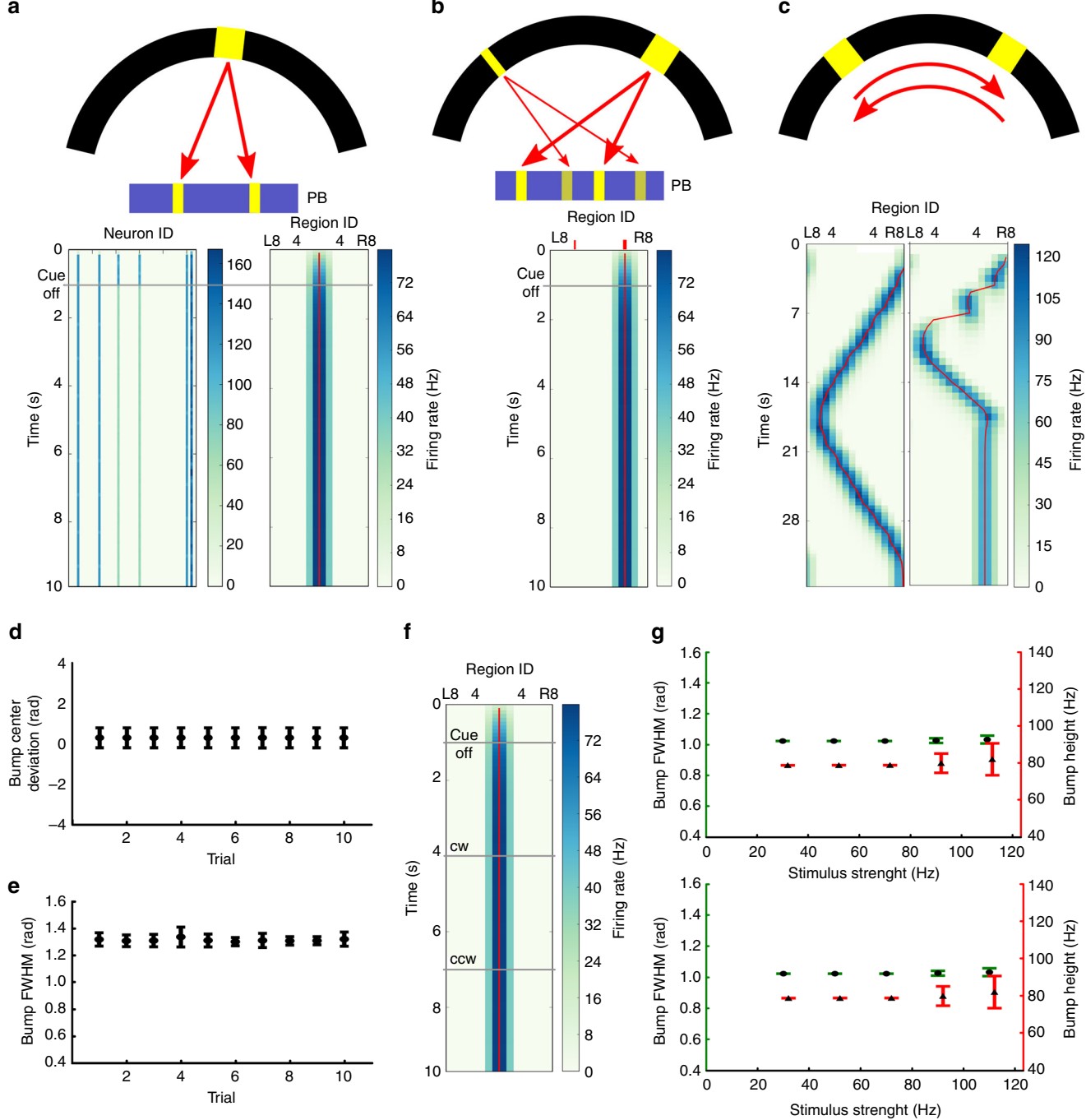

**Fig. 3** Persistent activity bump in the C-ring. **a** *Top*: A visual cue (*yellow bar* in the *black visual field*) is modeled by the spike input to two corresponding PB regions (see Methods). *Bottom left*: The visual cue elicited firing activity (represented by *graded cyan*) in several neurons and the responses persisted after the cue offset. *Bottom right*: A stable activity bump can be observed in the EB region-based firing rate plot. The *red line* is the estimated center of the activity bump, determined by the peak position of the Gaussian curve fit to the activity bump in each time step. **b** When we presented two visual stimuli with different brightnesses (*top*), the C-ring circuit developed only one activity bump in the EB (*bottom*) due to the nature of the neural competition. The thick and thin *red ticks* mark the EB regions corresponding to the stronger and weaker visual cues, respectively. **c** We investigated the response of the C-ring to a visual cue that moves across the visual field (*top*). *Bottom left*: the cue first moved clockwise for 16 s with a speed of 22.5° s⁻¹ and then reversed direction. *Bottom right*: the cue moved with a double speed for 8 s in each direction and then stopped. The circuit was able to accurately track the moving visual cue at a speed of 22.5° s⁻¹. The quality of the activity bump and its ability to track the moving cue (at 22.5° s⁻¹) are indicated by **d** the deviation between the bump center and the cue location, and by **e** the bump width (FWHM). The error bars indicate the within-trial s.d. **f** The activity bump is stable and does not shift in the dark during simulated body rotation (CW = clockwise, CCW = counterclockwise). **g** The height (*red*) and width (FWHM) (*green*) of the bump following stimulus offset were independent of the strength (*top*) and duration (*bottom*) of the stimulus, suggesting that the activity bump only encodes the direction of the stimulus, not its other non-spatial properties. Each data point was averaged over 10 trials and the error bars indicate s.d

**C-ring as a symmetric feedback circuit**. The C-ring circuit (formed by neurons of the PEI and EIP classes) constitutes a "symmetric" feedback circuit in which a connection from neuron A of the EIP class to neuron B of the PEI class is always accompanied by a reverse connection from B to A (Fig. 2c). This symmetric circuit exhibits a pattern of feedback excitation, which is widely proposed in theories of short-term/working memory[12, 37].

**P-ring as an asymmetric shifter circuit**. In contrast, EIP and PEN neurons do not form a symmetric feedback circuit in the P-ring. A close inspection of the novel representation depicted in Fig. 2d revealed that the P-ring constitutes a shifter circuit characterized by asymmetric feedback connections in which a connection from neuron A in the EIP class to neuron B in the PEN class is always accompanied by a connection of neuron B to a different EIP neuron (Fig. 2d). In fact, the EIP–PEN circuit can be separated into two subcircuits that represent different patterns of asymmetric connections. The first subcircuit (the inner gray circle in Fig. 2d) is characterized by clockwise transmission of signals along neurons in the EB regions. For example, EIP5$\xrightarrow{PB}$PEN5$\xrightarrow{EB}$EIP6$\xrightarrow{PB}$PEN6$\xrightarrow{EB}$EIP7, and so on. The label above each arrow indicates the neuropil where the synaptic connection occurs. In contrast, the second subcircuit (the outer *gray circle* in Fig. 2d) is characterized by counterclockwise transmission of signals along neurons in the EB regions. For example, EIP12$\xrightarrow{PB}$PEN10$\xrightarrow{EB}$EIP11$\xrightarrow{PB}$PEN9$\xrightarrow{EB}$EIP10, and so on. The clock-wise subcircuit includes neurons from EIP0 to EIP7 and from PEN0 to PEN7, while the counterclockwise subcircuit includes EIP10–EIP17 and PEN8–PEN15 neurons. Such asymmetric connection patterns suggest that any localized neural activity in the first or second subcircuits will shift clockwise or counter-clockwise, respectively. Interestingly, although the two subcircuits are coupled (as represented by *black* and *green arrows* in Fig. 2d) in the EB, the involved PEN neurons are separated and dis-tributed in two different hemispheres of the PB (*inner* and *outer rings* in Fig. 2d).

Based on the unique patterns of circuitry in the C and P rings, we postulated that the C-ring circuit is responsible for maintaining a stable activity bump, which represents the orientation of the salient visual cue during its presentation as well as following its offset, while the P-ring circuit shifts the bump in a direction opposite that of body movement (horizontal rotation) in the absence of all stimuli (Fig. 1b). We then tested this hypothesis by constructing a computational model with spiking neurons for the EB–PB circuit and systematically testing its ability in maintaining and shifting an activity bump.

**Persistent activity bump in the C-ring**. We first tested the function of the C-ring circuit model, which consists of the EIP, PEI, and R$_{EIP}$ neurons (ring neurons that make connections with EIP neurons in EB, see Methods). In the first task condition (condition 1 in Fig. 1e), we tested the C-ring circuit using a single visual cue with an arbitrary direction of orientation. A stable activity bump was observed in the EB regions corresponding to the direction of the visual cue, and the bump persisted even after the visual cue had been switched off (Fig. 3a). The full width at half maximum (FWHM) of the bump was 58.4° (1.02 rad). We further tested the model circuit by simultaneously presenting two visual cues in two different locations. Consistent with results obtained via empirical observation[10], the model circuit developed a single stable activity bump only in the regions corresponding to the direction of the most salient cue (Fig. 3b).

In the second task condition (condition 2 in Fig. 1e), we examined the response of the model to a moving visual cue (at a speed of 22.5° s$^{-1}$, or one EB region per second) and observed that the C-ring responded by moving the activity bump accordingly (Fig. 3c). We noted that the ability of the C-ring to track a moving cue depends on its speed. When a cue moves to a new position, it elicits a new activity bump through the local feedback excitation while it suppresses the old bump at the previous position through global feedback inhibition. This process will not have enough time to complete if the cue moves too fast. Our test showed that the C-ring circuit cannot track a moving cue with a speed faster than 45° s$^{-1}$, or two EB regions per second. At this speed, the circuit frequently lost track of the cue (Fig. 3c). Next, we measured the mean bump width as well as the deviation between the bump center and the cue location across several trials with a cue moving speed of 22.5° s$^{-1}$, observing that the activity bump closely followed the movement of the cue, with a FWHM of approximately 1.2 rad, which is comparable to results obtained via empirical observation (between $\pi/3$ and $\pi/2$ with large fly-to-fly variability)[10] (Figs. 3d, e).

In the third task condition (condition 3 in Fig. 1e), we examined whether the activity bump can be shifted by simulated body rotation after stimulus offset, a function we hypothesized for the P-ring. As explained in detail in Methods, the body rotation was simulated by adding unilateral excitation to one side of the PB (R0–R8 or L0–L8). We found that such an unilateral input did not shift the activity bump at all (Fig. 3f).

To ensure that the activity bump indeed only encodes the cue location, we further tested how the activity bump responds to other non-spatial properties (e.g., brightness or presentation duration) of the visual cue. An activity bump can be represented according to three properties: location, height, and width. We measured the height and width of the activity bump during the cue-off period and observed that both were independent of the brightness, and presentation duration of the visual cue (Fig. 3g), suggesting that the activity bump only encodes directional information associated with the visual cue and represents a suitable neural mechanism for encoding information regarding spatial orientation.

**Bump shifting in the P-ring**. We next tested the model of P-ring circuitry in the EB, which consisted of the EIP, PEN and R$_{PEI}$ neurons. Because the P-ring circuit also consists of feedback excitation loops and global feedback inhibition (via R$_{EIP}$ neu-rons), it was able to sustain an activity bump after stimulus offset (Fig. 4a), to maintain only one activity bump in response to two visual cues (Fig. 4b), and to closely follow the moving cue (Fig. 4c). However, the activity bump was less stable than that in the C-ring, as evident by the occasional drift (Figs. 4a, b). In the task condition 2, the trial averaged ($n = 10$) bump center devia-tion for the P-ring ($0.735 \pm 0.009$ rad) was significantly larger than that for the C-ring ($0.290 \pm 0.0005$ rad) ($p < 0.0001$) (Fig. 4d). Moreover, the trial averaged ($n = 10$) bump width for the P-ring ($1.436 \pm 0.009$) was also significantly larger than that for the C-ring ($1.216 \pm 0.013$ rad) ($p < 0.0001$) (Fig. 4e). The instability was likely due to the lateral excitation provided by the asymmetric connections from each EIP neuron to the PEN neurons that innervate neighboring EB regions (Fig. 2d). The result indicates that the P-ring circuit is less ideal in maintaining a stable activity bump than the C-ring circuit does.

Based on our result, we postulated that the function of the P-ring is to track rotation of the body in darkness in order to maintain spatial orientation. To accomplish this feat, the P-ring circuitry must perform angular path integration and shift the activity bump in the direction opposite that of body rotation. Analysis of the aforementioned asymmetric subcircuits in the P-

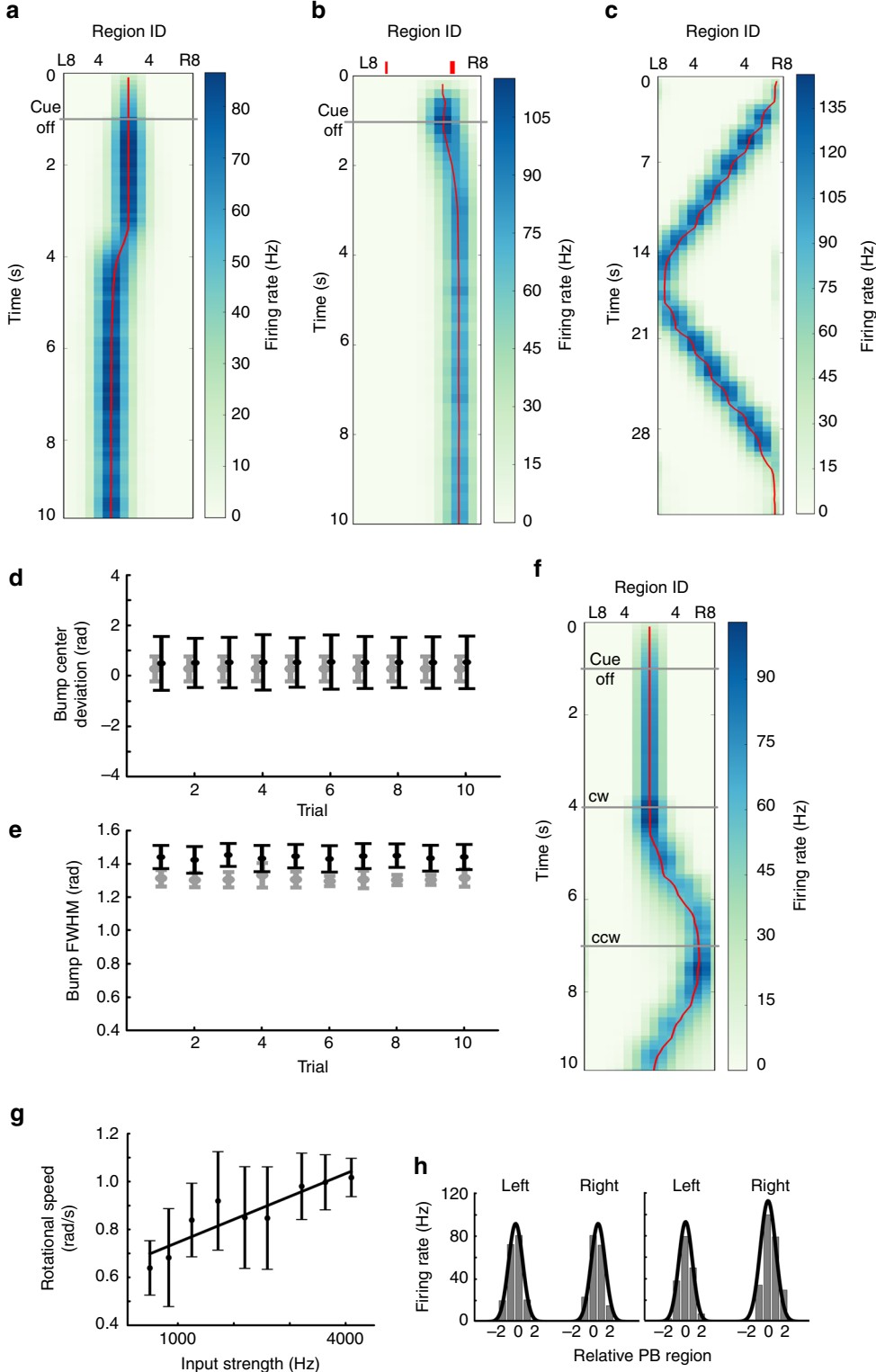

ring (Fig. 2d) suggests that, if we stimulate one side of the PB, we can activate the corresponding subcircuit and hence move the activity bump in the corresponding direction. Therefore, we simulated body rotation by addition of unilateral input with subthreshold excitation to one side of the PB (see Methods). The biological substrates underlying this unilateral input are subsequently analyzed in the Discussion. As predicted, our simulations showed that the activity bump moved clockwise or counterclockwise when the unilateral input was applied to the right (R0–R8) or left side (L0–L8) of the PB, respectively (Fig. 4f). To investigate the correlation between the strength of the unilateral input and the moving speed of the bump, we measured the mean moving speed of the bump in a 2-s trial for a given input strength and then calculated the averaged value over 10 trials. We observed that the moving speed of the bump is positively correlated with the strength of the unilateral input in a certain range (Fig. 4g).

Because each EB dual region exhibits major EIP projections to two contralateral PB regions, activity bumps in the EB result in bilateral PB activity bumps of the same size (height and width) (Fig. 4h, left). Furthermore, the model predicts that the bilateral activity bump in the PB will become asymmetric when a fly rotates its body horizontally (Fig. 4h, right). Such asymmetry occurs due to the unilateral input associated with rotation of the body, which in turn increases the sensitivity of the PEN neurons on one side of the PB to the input from the EIP neurons.

**Performance of the full EB–PB circuit model**. We next combined the two rings to construct a full EB–PB circuit model containing all three (EIP, PEI, and PEN) neuron classes. The goal was to create a model that combines the functions of both ring circuits for more realistic movement conditions, which consist of alternate forward walking and turning in darkness. Forward walking is characterized by steady orientation and hence requires the function of the C-ring circuit to maintain a stable activity bump, while the P-ring circuit is needed for turning in darkness because the activity bump has to shift in order to update orientation. In other words, the C and P rings must operate in coordination with one another in different movement states. To this end, we include two additional types of ring neurons, $R_{PEI}$ and $R_{PEN}$, which are alternatively activated during body rotation and forward walking, respectively (see Methods). Activation of $R_{PEI}$-ring neurons inhibits PEI neurons (the C-ring), allowing the PEN neurons (the P-ring circuit) to operate alone, whereas activation of $R_{PEN}$-ring neurons inhibits the P-ring but leaves C-ring activated. $R_{EIP}$-ring neurons, in contrast, need to be activated throughout the trial in order to sustain the activity bump (Fig. 5a).

Before we test the full model with a more realistic movement condition, it is informative to visualize the impact of each ring-neuron type on the network dynamics by adding them one by one to the circuit in a simple behavior task that consists of three periods of movement: forward walking, clockwise, and counter clockwise rotation (Fig. 5a). At first, we began with only the C and P rings (EIP, PEI, and PEI classes) without any ring neuron. With this condition, no activity bump could be formed (Fig. 5b). Next, we added each of the three ring-neuron types into the circuit and verified that the activity bump was only formed when the $R_{EIP}$ neurons were included in the circuit (Figs. 5c–e). However, the bump was not shifted by body rotation due to the persistently activated C-ring. Inclusion of either $R_{PEI}$ or $R_{PEN}$ did not work at all because no activity bump would be formed without $R_{EIP}$.

When all three ring-neuron types were included, the full model was able to form a stable activity bump, which shifted accurately with body rotation (Fig. 5f). We also tested whether the model works with only two of the three ring-neuron types. To this end, we performed a simulated "lesion" study, in which we suppressed each of the ring-neuron types individually. We injected a hyperpolarized current (−0.5 nA) to silence the specific ring neurons and effectively removed them from the circuit. We observed that suppression of $R_{EIP}$ neurons eliminated the activity bump entirely as expected (Fig. 5g). In contrast, lesions of $R_{PEI}$ neurons produced a strong yet immobile activity bump due to the continuous activation of the C-ring (Fig. 5h). Perhaps most interestingly, lesions of $R_{PEN}$ neurons resulted in continuous activation of the P-ring circuit, creating a broad activity bump that shifted with body rotation (Fig. 5i). In this case, the fly was still able to maintain spatial orientation, but with less accuracy due to the broad bump.

To test the full EB–PB circuit model under more realistic conditions, we created a random walk pattern consisting of a sequence of randomly interleaved right turns, left turns, and periods of forward walking (see Methods). Based on this random walk, we derived the corresponding sequence of ring-neuron activation and unilateral input to the PB, and subsequently applied the input sequence to the full EB–PB model (Fig. 6a), observing that the model circuit functioned appropriately (Figs. 6b, c). We further investigated how well the fly's own sense of rotation (as indicated by the position of the activity bump) in darkness represents the actual rotation of the body and discovered that, while the location of the activity bump initially aligned quite well with body rotation, the bump gradually drifted away from the actual orientation of the body (Fig. 6d)—a phenomenon also observed experimentally[10].

We further asked whether the full model is required at all for maintaining spatial orientation because P-ring circuit alone was able to sustain an activity bump and to shift it with body rotation (Fig. 4f). We tested the P-ring circuit and the full model using the random-walk task and found that the mean deviation between the bump location and the actual orientation of the body increased much more rapidly with the P-ring circuit than with the full model (Fig. 6e). The result suggests that the combination of the two rings and the three ring-neuron types greatly improves the fly's orientation memory. To visualize the trend, a square-root function, $y = \sqrt{ax}$, was fit to the data ($a = 0.049$ for the full model and $a = 0.138$ for the P-ring circuit).

It has been suggested that NMDA receptors play a crucial role in sustaining an activity bump (or working memory) in recurrent circuits[12, 13]. Therefore, we investigated the impact of the kinetic properties of the NMDA receptors on the dynamics of the model circuit. Compared to other types of excitatory receptors, NMDA receptors have three unique kinetic properties: long time constant (long-lasting synaptic current), magnesium block (voltage-dependent gating), and response saturation (limiting synaptic current at high-frequency input). We altered each of the properties and tested how they affect the activity bump. First, we tested a wide range of time constants and found that the model circuit was able to work with a smaller time constant down to roughly 50 ms (Fig. 7a, left). The model was unable to maintain a stable bump when the time constant is much smaller than 50 ms (Fig. 7a, right). The result is consistent with the notion that a long

**Fig. 4** Bump shifting in the P-ring. **a–f** Same as Figs. 3a–f. Like the C-ring circuit, the P-ring circuit was able to **a** develop an activity bump in response to a visual cue; **b** to maintain only one bump when presented with two cues (indicated by the *red ticks*); and **c** to follow the moving visual cue. However, the bump was less stable than that of the C-ring, as indicated by the occasional drift (**a**, **b**), by the larger variability in the bump deviation (**d**) and by the wider bump width **e**. In **d** and **e**, the data of the P-ring (*black*) are displayed together with those of the C-ring (*gray*, from Figs. 3d, e) for comparison. The error bars indicate the within-trial s.d. **f** The most distinct difference between the two rings was in their responses to the simulated body rotation in the dark. The activity bump shifted in the P-ring circuit but not in the C-ring circuit (Fig. 3f). **g** The trial-averaged moving speed of the bump was positively correlated with the strength of the unilateral input to PB. The error bars indicate s.d., while the straight line is the linear regression of the data points. **h** *Left*: The PB developed two activity bumps (*gray bars*, trial-averaged firing rate of the PEN neurons that innervate the corresponding PB regions) of the same height in response to the visual input. *Right*: The model predicted asymmetric bumps (*gray bars*) in the PB during horizontal body rotation in darkness due to unilateral input to the PB. The *black curves* in both panels depict the Gaussian fit to the activity bumps. The x-axis represents the EB region index relative to the peak of the bump on each side

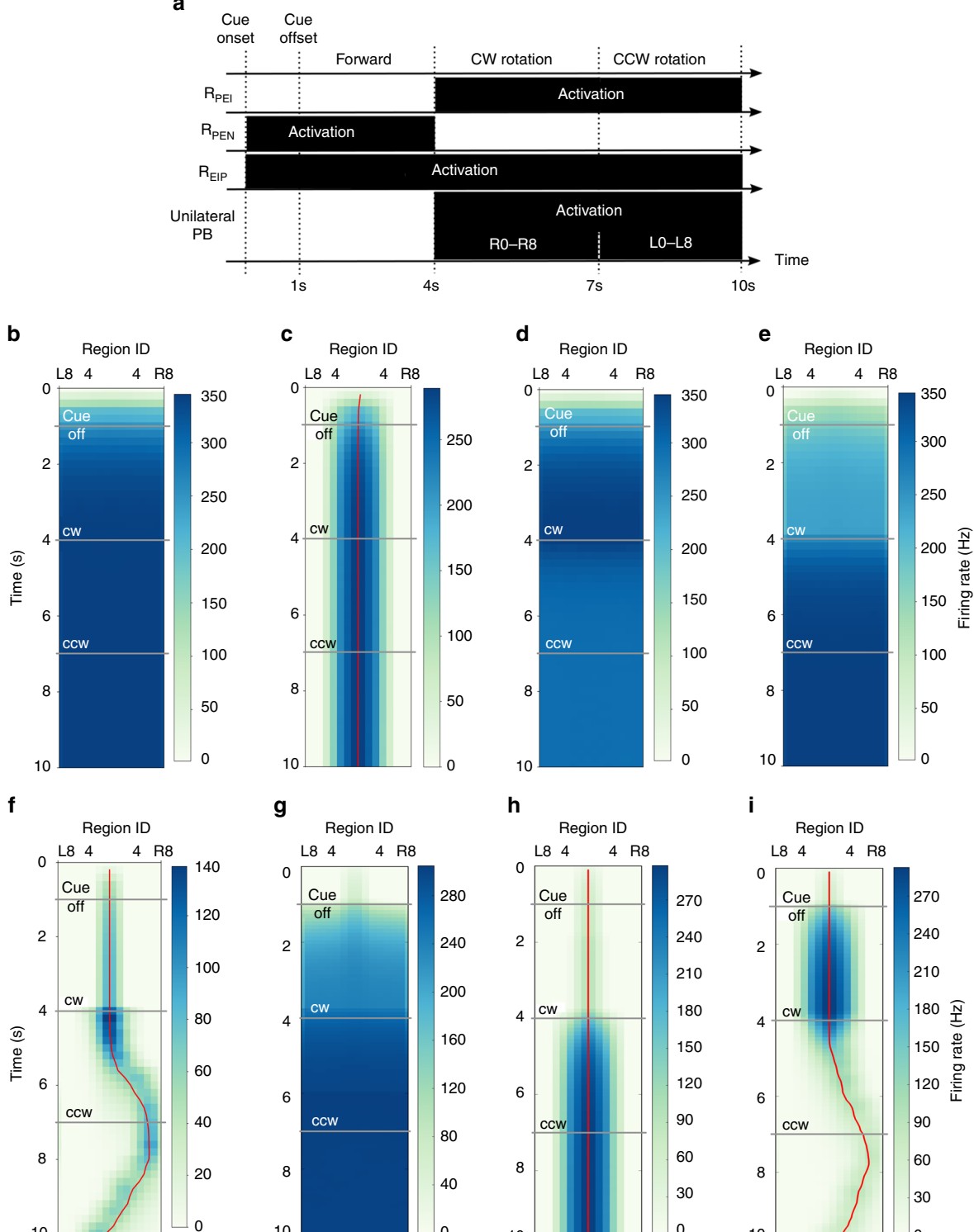

**Fig. 5** Maintenance of spatial orientation memory requires coordinated operation of the ring neurons in the full EP–PB model. **a** The full model was tested using a task consisted of four periods in the following order: cue-onset, cue-offset/forward walking, clockwise, and counter clockwise body rotation. $R_{PEN}$ ring neurons activate (to inhibit the P-ring) during the cue-onset and cue-offset periods, while $R_{PEI}$ ring neurons activate (to inhibit the C-ring) during both rotation periods. $R_{EIP}$ needs to be activated throughout the entire period of the task in order to maintain the activity bump, while the unilateral PB input is responsible for shifting the bump during the rotation periods. We tested the functions of the ring neurons by including them one at a time and demonstrated the result in the region-based firing rate plots for the EB (as in Fig. 3a) with the following conditions: **b** no ring neuron, **c** with $R_{EIP}$ neurons only, **d** with $R_{PEI}$ neurons only and **e** with $R_{PEN}$ neurons only. We also tested the full model by taking out each ring-neuron type one at a time from the model with the following conditions: **f** the full model, **g** $R_{EIP}$ lesion, **h** $R_{PEI}$ lesion and **i** $R_{PEN}$ lesion. The result indicates that only when all ring neurons function properly (**f**), the model circuit is able to maintain a sharp activity bump that encodes the spatial orientation memory. Although a functioning activity bump was also formed in the condition of $R_{PEN}$ neurons lesioned (**i**), the activity bump was too broad. In **b–i**, the red line is the estimated center of the activity bump

synaptic time constant is essential in maintaining working memory (in the form of persistent neural activity) in a recurrent network[13, 14]. Second, we removed magnesium block by setting $[Mg^{2+}]$ equal to 0 in Equation 1. We found that the model circuit was still able to maintain an activity bump (Fig. 7b). Finally, we removed the mechanism of response saturation by removing the term $(1-S(t))$ from the right-hand side of Equation 2. The removal greatly increased the synaptic current, hence we had to scale down the synaptic weight for compensation. We multiplied the weight by a factor and found that the activity bump could only be observed when the factor was between 0.08 and 0.09. However, the bump completely disappeared after the body rotation initiated (Fig. 7c).

To complete our test, we examined how the weight of the NMDA synapses may affect the dynamics of the model circuit. We measured the mean deviation of the bump position and the actual body orientation and found that the model tracked the body rotation reasonably well in a moderate range between 100 and 120% of the original weight (Fig. 7d). It is also important to ask whether the weight of cholinergic synapses, the other type of excitatory synapses implemented in the model, is also crucial to the model performance. We found that it did not change significantly across a broad range of the synaptic weight (between 70 and 150% of the original value) (Fig. 7e). Note that cholinergic synapses were only implemented in the visual input and the unilateral input to the PB and do not directly interact with the internal dynamics of the model circuit. Therefore, the insensitivity of the model performance to the weight of the cholinergic synapses is expected. On the other hand, the result suggests that the model circuit is robust against changes in the input strength.

## Discussion

In the present study, we constructed the connectome, which describes the EB–PB circuitry in the central complex of the fruit fly based on recently published anatomical data[32, 33], and discovered symmetric and asymmetric feedback circuits in the system. We constructed a spiking neural network model based on the connectome at the single-neuron level, and demonstrated that the symmetric circuit is capable of sustaining a stable activity bump, while the asymmetric circuit provides a mechanism for shifting the activity bump. The simulated neural activity reproduced empirical observations[10], suggesting that the proposed model provides a biologically plausible mechanism for maintaining spatial orientation and performing angular path integration.

The mechanism underlying angular path integration has also been addressed in a number of computational models of rodent head-direction system[19, 24–29]. However, these models have been proposed on a more abstract level without the availability of single-cell-level connectomes. A majority of the these models has been developed based on the idea that the activity bump is maintained by attractor ring networks and the bump is shifted by the offset connections (the shifter circuits) between rings[19, 25, 26, 29]. In contrast, the EB–PB system implements the shifter circuit between a linear (PB) and a circular (EB) structure (Fig. 8a). The PEI neurons innervate eight regions on each side of the PB (R0–R7 and L0–L7) and form perfect one-to-one feedback circuits with the eight dual region of the EB through the EIP neurons (Fig. 8b). To accommodate the shifted innervation of the PEN neurons, two additional regions (R8 and L8) have to be added to the PB. Furthermore, in order to complete the shifter circuit, two more EIP neurons (types 0 and 17) that innervate the PB R8 and L8 regions also need to be added (Fig. 8c). This explains why there are nine PB regions and nine EIP neuron

types, but only eight PEN and eight PEI neuron types in each side of the fly brain.

The proposed model is able to provide specific and experimentally testable predictions. First, the model predicts that when a fly horizontally rotates its body, the bilateral activity bumps in the PB become asymmetric (Fig. 4h). Second, the model predicts the distinct roles of PEN and PEI neurons, as well as those of the ring neurons (Fig. 5), in spatial orientation. These predictions may be easily verified by calcium imaging or by optogenetics if proper drivers for the targeted neuron types are available.

In our model, the activity bump can be shifted when unilateral input is provided to only one side of the PB. Although this finding provides a mechanism for updating spatial orientation in darkness, precisely how the unilateral input is generated and associated with rotational movement remains unclear. One possibility is that the unilateral input to the PB is elicited by the efference copy or a reafferent signal of the rotation command produced in downstream motor regions. Indeed, a larger-scale connectome analysis[38] revealed heavy inputs to the PB from the DMP (dorsomedial protocerebrum), a known motor region, as well as from the CCP and VMP (Supplementary Table 4a). Activation of the left (or right) regions of the CCP or VMP may lead to activation of the entire right (or left) side of the PB. Interestingly, both CCP and VMP receive a strong input from the motor region DMP (Supplementary Tables 4b and c). Another possibility of the source of the body rotation signal, at least during flight, is the halteres, which are organs that sense inertia[39–41]. Put together, we postulate that the entire system functions as a neural integrator[42, 43] that performs double integration. The aforementioned structures provided information about rotational acceleration and is transformed into angular velocity by the first integration. The velocity signal is then fed into the PB as an unilateral input, which is further transformed into the rotational angle by the second integration performed in the EB–PB circuit. We noted that the correct synaptic strength for the unilateral input is required by the EB–PB circuit in order to accurately perform the second integration. Incorrect synaptic strength causes mismatch between the body rotation and the bump movement. We conjecture that the correct synaptic strength can be achieved by some form of plastic changes in synapses that occur during the development of the central complex, which start in the third instar larval stage and mature after metamorphosis[44].

It has been found that the EB does not have a fixed retinotopic mapping of the surrounding; the mapping changes between individuals and even across trials in the same fly[10]. In the proposed model we do not consider such a variable mapping and it is very interesting to address it in a follow-up study. One plausible mechanism is that the activity bump is spontaneously produced at a random location on the ring. The visual system does not provide a retinotopic input to the circuits but rather provides the motion speed signal, as hypothesized by an independent study published recently[31]. This signal is then transformed into the unilateral input to PB and hence shifts the bump in the same way that the hypothesized haltere signal or the motor command feedback does. However, this mechanism suggests that the activity bump starts at a random location in the same fly in each trial, which is not entirely consistent with observations[10]. Therefore, some degree of retinotopic input to the EB–PB circuits may still exist.

The proposed model can be expanded in the future to address the following interesting questions:

1. How is a straight path integrated in the fly brain? The navigation system needs to take signals from angular and straight path integrators in order to properly orient in a landscape. A recent study on central complex's responses to progressive and regressive optical flows[45] may provide

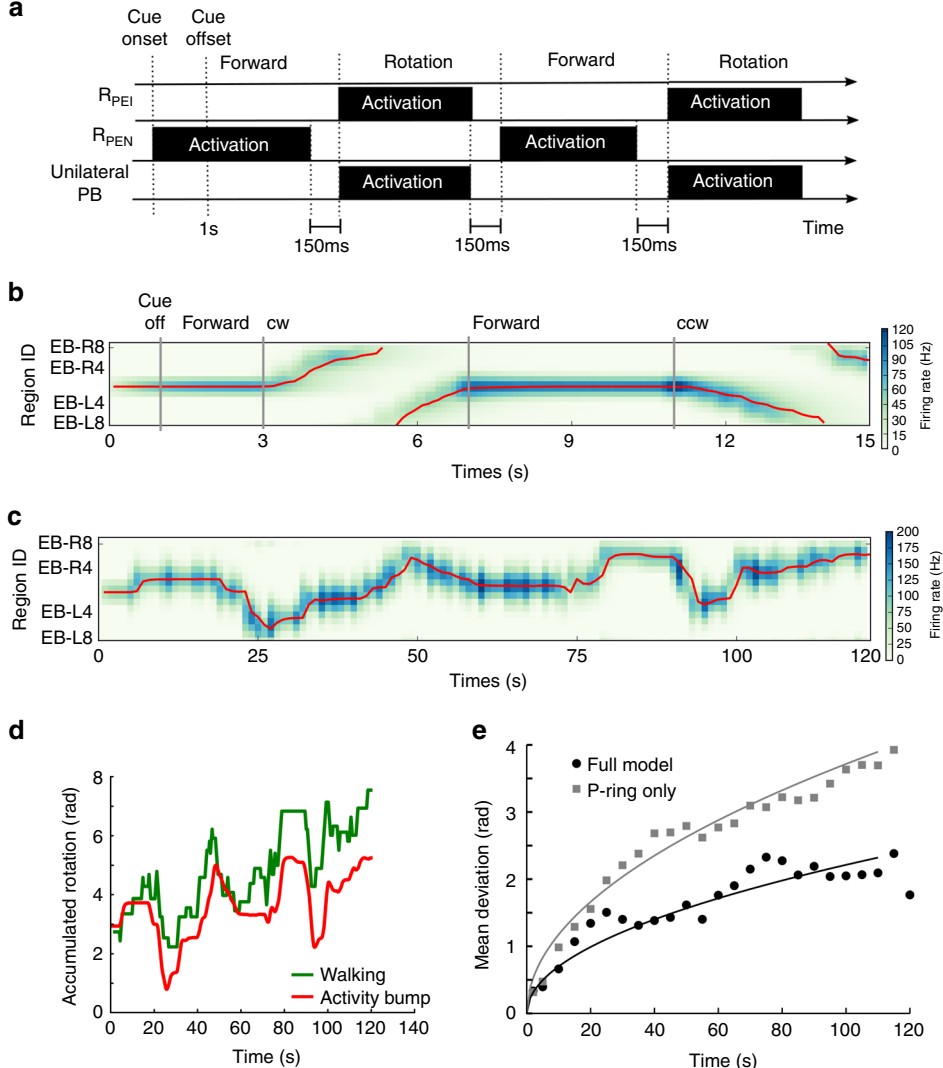

**Fig. 6** The maintenance of spatial orientation memory in the full EB–PB circuit in a random walk paradigm. **a** A schematic of the input protocol for a random walk. A random walk consists of a sequence of interleaved periods of forward walking and rotation. During forward walking the $R_{PEN}$ ring neurons are activated to inhibit the P-ring, while during the rotation the $R_{PEI}$ ring neurons are activated to inhibit the C-ring. The PB receives a unilateral input when the simulated fly performs a rotational movement. $R_{EIP}$ neurons (not shown) are activated throughout the entire trial. To allow smooth transitions between forward walking and rotation, each input has to cease 150 ms prior to the end of the corresponding movement type (see Methods). **b** As shown in the region-based firing rate plot, the full EB–PB circuit produced a stable activity bump that continuously updated its position when the body rotated in darkness. **c** EB activity of an example trial of a full period (120 s) of random walk in darkness (cue off at $t = 1$ s). **d** The perception of spatial orientation, as indicated by the peak position of the activity bump in EB (*red*) closely reflected actual fly body orientation (*red*), though the deviation between the two increased with time. **e** Trial-averaged ($n = 20$) deviation between the perception and actual body orientation increased with time (*black circles*: the full model, *gray squares*: the P-ring-only model). To visualize the trends of the data, we fit a square-root function to the data as represented by the curves. The result indicates that although the P-ring itself can track the changes of orientation, the accuracy is much worse than that of the full EB–PB model

insights into this issue. On the other hand, the efference copy of forward or backward movement commands should also play an important role, in particular when a fly moves in the dark. We conjecture that fan-shape body (FB) and/or associated neuropils may take part in straight path integrations, and the downstream circuits may be where a "cognitive map" is maintained. A number of theories or network models of spatial navigation and the cognitive map have been proposed for rodents[46, 47] and insects[30, 48], and may provide insight into this question.

2.  How do visual feature selectivity and visual pattern memory play roles in spatial orientation? A previous study has reported that the R2 and R4d ring neurons, which respectively innervate the A and O rings of the EB, exhibit complex selectivity to spatial parameters such as bar orientation, azimuth, and elevation angles of visual objects[9]. Moreover, R2/R4m neurons and some FB neurons have also been shown to be associated with the memory of different visual patterns[49, 50]. These circuits may interact with the C- and P-ring circuits and modulate spatial orientation.

3.  How does the system maintain spatial orientation during flight in three-dimensional space? Several studies have shown that the central complex and associated regions exhibit distinct neuronal responses between flight and walk[9, 45, 51, 52]. Therefore, it is interesting to study this question by extending the current model beyond the PB and EB.

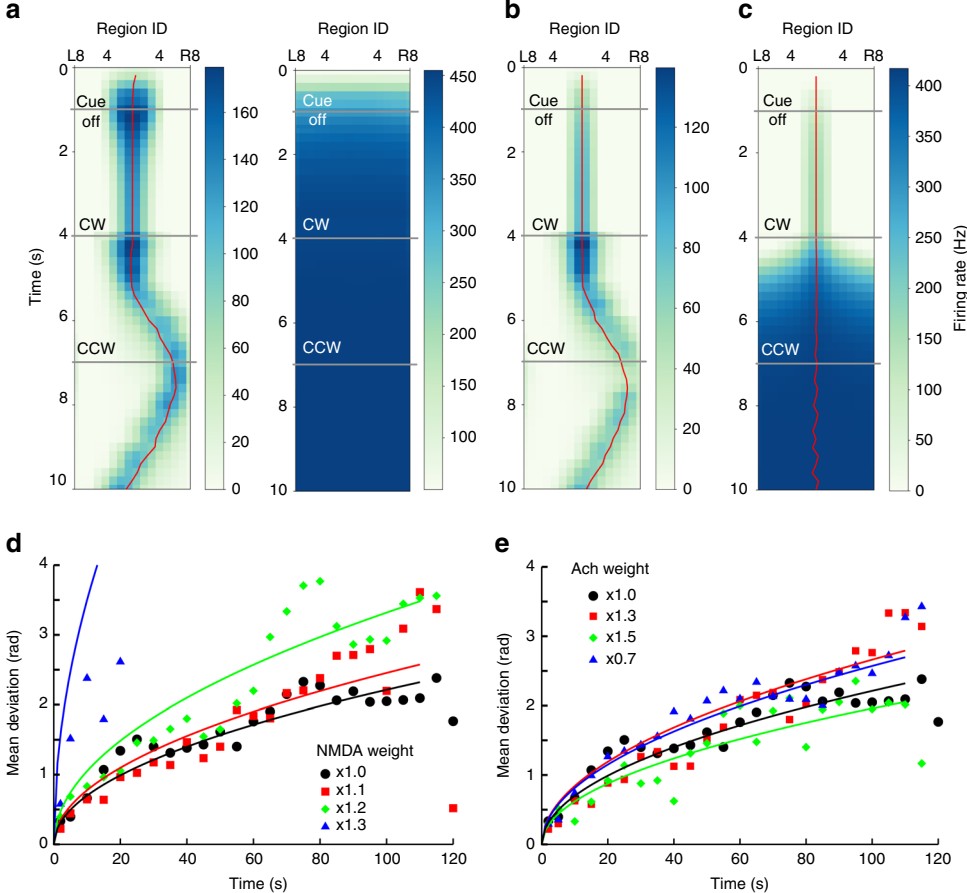

**Fig. 7** The impact of synaptic kinetics on spatial orientation memory in the full model. The NMDA receptors play a crucial role in the dynamics of the model. We tested how NMDA-specific kinetics influences the maintenance of the activity bump using the same task as in Figs. 5b–i and demonstrated the result in region-based firing rate plots. **a** The model circuit worked with a synaptic time constant as small as 50 ms (*left*) but failed to produce any activity bump with a much smaller time constant such as 10 ms (*right*). Note that reducing the time constant leads to a smaller synaptic current; hence we needed to compensate the reduced current by scaling up the synaptic weight. The best scaling factor was found to be 1.2 when the time constant equals 50 ms. **b** We removed the mechanism of magnesium block from NMDA and found that the removal produced little impact to the activity bump. **c** We turned off the saturation mechanism and discovered that, although a bump presented initially, it quickly disappeared after the initiation of body rotation. **d** We tested the model with different NMDA synaptic weights in the random walk task and measured the mean deviation (as in Fig. 6e). Each color denotes data for the mean deviation produced by the specified synaptic strength, e.g., ×1.3 for 130% of the original strength used in Fig. 6. The model worked reasonably well within a moderate range (×1.0–×1.2) of the NMDA synaptic strength. **e** For comparison, we further tested how the strength of the cholinergic synapse (Ach) influences the model performance and found that the model was insensitive to the change of the synaptic weight. To visualize the data in **d** and **e**, we fit the data to a square-root function as represented by the curves

In conclusion, we proposed a neural circuit model for the maintenance of spatial orientation and angular path integration. The present work is significant in several aspects. First, the study is one of the few examples of neural circuit models built upon detailed connectome analysis down to the single-neuron level. Second, the present connectome supports the notion of asymmetric rings proposed in theoretical studies, although the actual circuit implementation is quite different. Third, with highly detailed neural circuits, the proposed model is able to make experimentally testable predictions at the level of single neurons.

## Methods

**Neuroanatomy of EB and PB**. We constructed our model of EB–PB circuits based on our previous analysis[34] and on data published in two recent studies[32, 33]. Lin et al.[32] described the detailed innervation pattern of each observed and predicted neuron type in the central complex. Wolff et al.[33] further provided updated information on the innervation patterns of several neuron types and described new patterns of segmentation in the PB. Chang et al.[34] analyzed the topographical mapping in the central complex network. Based on these papers, our circuit included the PB (18 regions/glomeruli, nine on each side), the EB (C and P rings, 16 regions/sectors/wedges in each ring), and four neuron classes: EIP, PEI, PEN,

and ring neurons (Figs. 1d, 2b, and Supplementary Table 1). In the present study, we defined two levels of morphological classification: class and type. Each neuron type represents a unique innervation pattern at the regional level, while a neuron class is a collection of neuron types with similar innervation patterns at the neuropil level. Neurons in the EIP class innervate both C and P rings with their dendritic domains in three consecutive regions of each ring, while they innervate only one region in the PB. Note that, as reported by Wolff et al.[33], EIP0 and EIP17 innervate only one EB region, EB R8 and L8, respectively. The reason why these two neuron types exhibit such atypical innervation patterns is unclear. But their existence does not affect the neural activity in EB R8 and L8 of the simulated EB–PB circuit. Neurons in the PEI class innervate one PB region with their dendritic domains, while they innervate two consecutive regions in the C-ring of the EB with their axonal domains. Neurons in the PEN class are similar to those in the PEI class but with axonal innervation in the P-ring of the EB. The ring neurons project from the lateral triangle (LT, or bulb) to all regions in the EB, and different ring-neuron types innervate different rings of the EB[32, 35, 36, 53].

Because EB and PB neurons are found to express the VGlu driver[54, 55], and because NMDA receptors in the EB play a crucial role during a memory task[56], part of the neuronal interaction in the EB–PB circuits may be mediated through NMDA receptors. Therefore, we hypothesized that the mutual excitation between EIP, PEI, and PEN is mediated through NMDA receptors. In addition, constituents of acetylcholine machinery are also widely expressed in EB and PB[32, 57], and in the projection neurons from upstream regions to the PB[32]. Therefore, we modeled the visual inputs with cholinergic synapses. We acknowledge the possibility that the

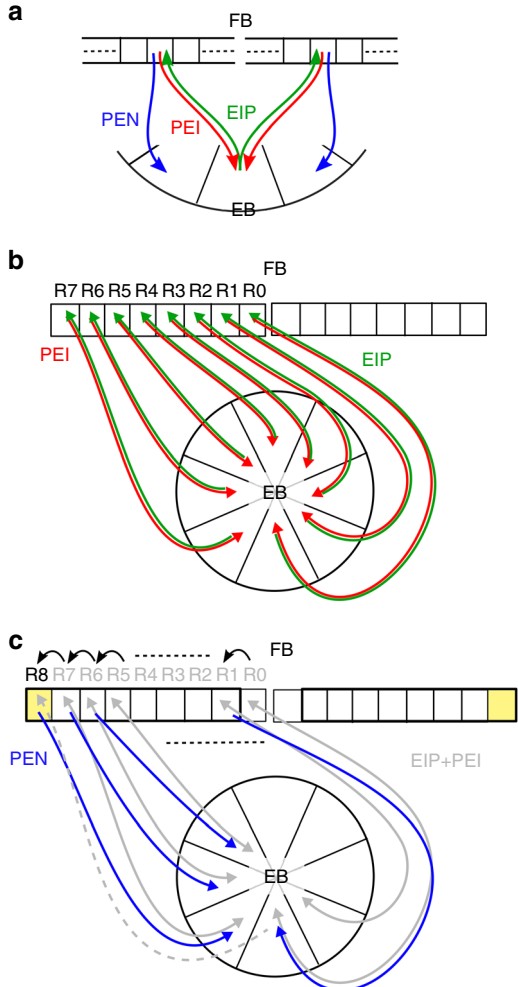

**Fig. 8** Schematics of the shifter circuit in the EB–PB system. **a** In the EB–PB system, the PEN neurons form a shifter circuit by sending the signals from the PEI–EIP feedback circuit to the neighbor EB regions. We illustrate how the shifter circuit is formed in the **b** and **c**. **b** Consider the PEI neurons which innervates R0–R7 regions. Each PEI neuron sends signals from one EB region to one PB dual region and the signals are sent back to the same PB region by an EIP neuron. Note that an EIP neuron actually innervates three EB regions, but two of them belong to the same dual region. For the simplicity, we only depict the innervation of the dual region for each EIP neuron. **c** To build a shifter circuit for the PEN neurons, we first copy the innervation patterns of the PEI neurons and then shift the innervation sites in PB rightward by one region but leave the innervation sites in EB unchanged. To accommodate the change, we need to add a new PB region, R8, to the right side of PB. A new EIP neuron (*gray dashed line*) that innervates R8 has also to be added in order to complete the shifter circuit. Note that here we only depict the neurons innervating the right side of PB. The neurons innervating the left side of PB are simply the mirror images of the neurons in the right side

EIP, PEI, and PEN neurons are cholinergic. But it does not affect the core computational principle we demonstrate here because cholinergic synapses are excitatory, and computationally they function similarly to glutamatergic synapses with AMPA or NMDA receptors. Based on a previous immunocytochemistry study[35], we modeled the ring neurons as GABAergic neurons.

We defined a "dual region" in the EB as a combination of two consecutive regions of the EB. For example, the dual region L23 represents the combination of regions of L2 and L3. This reduced the 16 regions of the EB to only eight dual regions and greatly simplified the network analysis and graphical representation. Such reduction is also anatomically meaningful because each PEN or PEI neuron innervates one such dual region in the EB (Figs. 2a, b).

**Neuron and synapse models**. In the proposed network model, each neuron is simulated by the leaky integrate-and-fire model with conductance-based synapses as described in previous studies[58–60]. The membrane potential $V_i$ of a neuron $i$ is given by

$$C_\mathrm{m}\frac{\mathrm{d}V_i}{\mathrm{d}t} = -g_L(V_i - V_L) - \sum_j I_{ij},$$

where $C_\mathrm{m}$ (=0.01 nF for $R_\mathrm{EIP}$ neurons and 0.1 nF for all other types of neurons) is the membrane capacitance, $g_L$ is the leak conductance and is set accordingly so that the membrane time constant $\tau_\mathrm{m}(= C_\mathrm{m}/g_L)$ equals 15 ms, $V_L$ (=−70 mV) is the resting potential, and $I_{ij}$ is the synaptic current elicited by spike input from neuron $j$. The spike threshold is set to −50 mV, and the reset potential is −70 mV. There is no direct measurement of these parameters for the central complex neurons. Therefore, we had to make assumptions about the parameters based on the commonly used values. The resting potential, reset potential, and the spike threshold are often similar between different neuron types. The membrane capacitance and time constant were set in accordance with those observed in the olfactory system of fruit flies[61, 62]. We stress that the exact values of these parameters are not crucial for the proposed model because our goal is not to reproduce accurate electrophysiological properties for individual neurons but rather to demonstrate the behavior of the circuits, which could work in a large range of parameter values.

The synaptic current is described by

$$I_{ij} = \sum_j g_{ij}s_{ij}(V_i - E)$$

where $g_{i,j}$ is the maximum synaptic conductance, $s_{ij}$ is the gating variable between neurons $i$ and $j$, and $E$ is the reversal potential of the given synaptic receptors. We modeled three types of receptors: GABA$_\mathrm{A}$, acetylcholine, and NMDA. The reversal potentials are 0, 0, and −70 mV for the acetylcholine, NMDA, and GABA$_\mathrm{A}$ receptors, respectively. The gating variable is described by

$$\frac{\mathrm{d}s(t)}{\mathrm{d}t} = -\frac{s}{\tau} + \sum_k \delta(t - t^k)$$

for the GABA$_\mathrm{A}$ and acetylcholine receptors, where $\tau$ is the time constant of the synapse (5 ms for GAB$_\mathrm{AA}$ and 20 ms for acetylcholine), $\delta$ is the delta function, and $t^k$ is the time of the $k$-th presynaptic spike. For the NMDA receptors the conductance $g_{ij}$ is voltage-dependent:

$$g_{ij} = G_{ji}/\left(1 + [\mathrm{Mg}^{2+}]\mathrm{e}^{-0.062V_i/3.57}\right), \qquad (1)$$

while the gating variable saturates under high-frequency input:

$$\frac{\mathrm{d}s(t)}{\mathrm{d}t} = -\frac{s}{\tau} + \alpha(1 - s(t))\sum_k \delta(t - t^k), \qquad (2)$$

where $[\mathrm{Mg}^{2+}]$ (=1.0 nM) is the extracellular concentration of $\mathrm{Mg}^{2+}$, $\tau$ (=100 ms) is the time constant of the channels, and $\alpha$ (=0.63) is a factor used to compensate the overshooting of the gating variable due to the neglect of the rise time in the model.

**The spiking neural network model of the EB–PB circuits**. We constructed a computational model with spiking neurons for the EB–PB circuit based on the connectome (Supplementary Table 2) we derived from the innervation data (Supplementary Table 1). Because the innervation data only provides information about innervated regions of each neuron, the connections between neurons were inferred according to the following basic assumption: a neuron that innervates a region of the EB or PB with an axonal domain makes a synaptic connection with neurons that innervate the same region with the dendritic domains. The assumption was made based on the observation that neurons innervating the same region usually have their arbors heavily overlapped in the region. The overlap between the dendritic and axonal arbors strongly suggests the existence of synapses[63–65].

A persistent activity bump requires not only strong local feedback/recurrent excitation but also global inhibition. Therefore, we included GABAergic ring neurons in our model. The GABAergic ring neurons are not part of the EB–PB circuit but project from the LT to the EB and form a ring-like arborization that infiltrates all regions (or wedges) in the EB[35, 36]. These neurons can be categorized into several types and each innervates different EB rings[32, 53]. In the present study, we model neurons that innervate the C-ring and P-ring. The C-ring corresponds to the R1 unit as defined in Renn et al.[53], while the P-ring is undefined in Renn et al.[53] but was described in Lin et al.[32] and in Wolff et al.[33]

Although the neural circuit was proposed based on observed data, to make a functioning circuit, several additional assumptions are required. First, the function of EB–PB circuits involves three types of ring neurons ($R_{EIP}$, $R_{PEI}$, $R_{PEN}$) whose inhibitory projections respectively extend to the EIP, PEI, and PEN neurons. Based on the anatomy described above, $R_{PEI}$ and $R_{PEN}$ correspond to R1- and P-ring neurons, respectively, while $R_{EIP}$ may correspond to subsets of R1- and P-ring neurons because EIP neurons innervate both C and P rings. Second, the $R_{EIP}$-ring neurons receive indirect excitatory projection from the EIP neurons and in turn provide feedback inhibition to EIP neurons. For simplicity, in the current model we allowed the $R_{EIP}$ neurons to receive direct input from the EIP neurons. The simplicity does not significantly change the dynamics of the model circuits. Third, the $R_{PEI}$- and $R_{PEN}$-ring neurons serve alternate functions. When a fly is not moving, its $R_{PEN}$-ring neurons are activated and inhibit the PEN neurons, allowing the C-ring circuit (PEI-EIP) to maintain a static activity bump. When the fly rotates horizontally, $R_{PEI}$-ring neurons are activated and inhibit the PEI neurons, allowing the P-ring circuit (PEN-EIP) to shift the activity bump.

The strengths of synaptic connections between neurons were regarded as free parameters and used to tune the dynamics of the neural circuits. Supplementary Table 3 includes the values for synaptic conductance between neuron classes. During the tuning procedure, we imposed one constraint: if the neurons in class A project to the neurons in class B, their synaptic strengths are proportional to the number of regions where the axonal and dendritic domains overlap. For example, the PEI11 neuron makes a connection with the EIP12 neuron in two EB regions (R2R3) (Fig. 2a). The synaptic strength of this connection is the same as that between PEI10 and EIP11, which also makes a connection in two EB regions, but is twice that of the PEI11 → EIP4 connection, which occurs in only one EB region (R2). There is one exception for EIP0 and EIP17, which exhibit an atypical innervation pattern by only innervating one instead of three EB regions. Because of this atypical innervation pattern, EIP0 and EIP17 receive connections from PEN8 and PEN7 in only one EB region. However, these connections are crucial for continuously shifting the activity bump when it reaches the dual region R8L8. In order to ensure smooth shifting of the activity bump, the synaptic strength of these connections was increased threefold (Supplementary Table 3).

In the simulations, each neuron type was simulated using a pool of 10 identical neurons. A connection between two neuron types was modeled as an all-to-all connection between the two pools. We stress that the exact number of neurons per pool is not essential to the model. The reasons why we assume multiple neurons in each neuron type are threefold: first, given the low firing rate (tens of spikes per second) observed in the central complex of some insets[5, 66, 67], we need multiple neurons per neuron type in order to produce firing rates that are large enough to provide sustained excitation or inhibition to the downstream neurons, second, by sampling spikes from a number of upstream neurons rather than from just one, a neuron can produce a much reliable response to a sensory stimulus, and third, it makes sense to have redundancy. If each neuron type consists of only one neuron, damage to a single neuron can easily impair the function of spatial orientation. To summarize, the first reason is required in order for the model to work and the rest two reasons are purely based on functional consideration.

**The spatial orientation task**. We tested our model under three types of spatial orientation task conditions: static cue-off, dynamic cue-on, and dynamic cue-off (Fig. 1e). In all three tasks, a trial began with the presentation of a visual cue for 1 s. The three task conditions differed after the 1-s cue period. In the static cue-off condition, the cue was turned off after 1 s. In the dynamic cue-on condition, the visual cue remained on, but shifted clockwise or counterclockwise with a speed of $22.5°\,s^{-1}$. In the dynamic cue-off condition, the visual cue was turned off after the 1-s cue period and followed by horizontal body rotations.

We mapped the 16 EB regions homogeneously to the 360° horizontal visual space. If a visual cue was presented in the visual field of an EB region, the cue was simulated by adding bilateral inputs (spike rate of 50 Hz through cholinergic receptors with a maximum conductance of 2.1 nS) to the two PB regions from which the PEI neurons projected to the given EB region. For example, if a visual cue was presented at 40°, which is in the receptive field of the EB region R2, the cue was simulated by sending the spike input to the R5 and L3 regions of the PB. Our implementation of the visual cue input through the PB is based on the following consideration. There are four suggested visual input pathways to the central complex[32]. Three of them enter the central complex through large-field inhibitory (GABAergic) or modulatory (dopaminergic) neurons, which are not likely to elicit a localized activity bump in the EB. The fourth pathway enters the PB through small-field cholinergic neurons, which can provide excitatory input to specific PB glomeruli. The input has to be injected into two bilateral PB regions because they converge to the same EB region symmetrically. Otherwise, a unilateral input to PB will cause a moving activity bump in the P-ring even when the visual cue is static. Another indirect support comes from the observation that in the locust the PB responds to the E-vector of the polarized light in a bilateral fashion, suggesting that the visual input to PB is organized bilaterally[5, 67]. We would like to emphasize that from the modeling point of view injecting the visual input either to two bilateral PB regions or to the corresponding EB region produces the same result (eliciting the same activity bump). Therefore, if future study identifies a new visual pathway to individual EB regions responsible for eliciting the activity bump, the finding will not affect the proposed circuit principle about spatial orientation.

**The simulated body rotation**. The asymmetric subcircuits in the P-ring circuit (Fig. 2d) revealed that the PEN neurons (PEN0–PEN7) in the clockwise subcircuit innervate the PB contralateral to the PEN neurons (PEN8–PEN15) in the counterclockwise subcircuit. Therefore, each subcircuit can be independently manipulated through the addition of an unilateral input to one side of the PB. This circuit arrangement provides a perfect mechanism for shifting the activity bump in the desired direction. Based on the observation, a counterclockwise body rotation was simulated by applying a unilateral input (spike rate of 3150 Hz through cholinergic receptors with a maximum conductance of 0.3 nS) to all PB regions in the right side (regions R0–R8), while a clockwise body rotation was simulated by applying a unilateral input of the same strength to the left side of the PB (regions L0–L8). Although the spike rate of the unilateral input appears to be large, the number is assumed to be the total spike input from a group of upstream neurons and hence is not biologically unrealistic.

As mentioned in the previous section, C-ring and P-ring circuits were assumed to function alternately when a fly switched between the forward walking and rotating states. When one ring is activated, the other must be inhibited. The inhibition was accomplished by applying a strong excitatory input (spike rate of 200 Hz through cholinergic receptors with a maximum conductance of 10 nS) to the corresponding ring neurons. However, when the switch between the ring neurons occurred exactly at the time of the change of the movement states (from forward walking to rotating or back), the activity bump occasionally disappeared. This is because the circuits were not fast enough to transfer the bump from the C-ring to the P-ring or back. To address this issue, we assumed that activation of the $R_{PEN}$ neurons ceases (implemented by removing the excitatory input) 150 ms prior to the initiation of body rotation (Fig. 6a), and that activation of the $R_{PEI}$ neurons ceases 150 ms prior to the end of body rotation. This protocol allows the two rings to co-activate shortly before the change of the movement state so that the activity bump is able to smoothly transfer between the two rings. Such an assumption is not quite arbitrary considering the preparatory neural activity observed in various premotor systems (primate saccadic eye movements, for example refs. [68, 69]).

**The simulated random walk**. To generate a random walk protocol for the purpose of testing the EB–PB circuit model, we established the relationship between the unilateral input to the PB and the rotation speed of the body, as this unilateral input directly represents bodily rotation. Observations from a previous study indicate that the activity bump in the EB matches the actual body orientation accurately in the first few seconds after beginning the walk in darkness[10]. Therefore, we performed the spatial orientation task under the dynamic cue-off condition. For a given PB unilateral input strength, we measured the speed of bump movement for the first 2 s after cue offset and calculated the average speed over 10 trials. Because the 16 EB regions were mapped to 360° of the visual field, the moving speed of the bump is easily converted into the body rotation speed. Finally, we repeated the procedure for different input strengths and established a relationship between the input strength and the simulated body rotation speed (Fig. 4g).

Next, we constructed a random walk for the simulated fruit fly. The random walk consisted of a sequence of interleaved periods of forward walking and rotation. Forward walking was simulated by the activation of $R_{PEN}$ without unilateral input to the PB, while rotation was simulated by the activation of $R_{PEI}$ with a unilateral input to PB (L0–L8 for a clockwise turn and R0–R8 for a counterclockwise turn). The length of each period of forward walking was randomly determined for a duration between 0.2 and 2 s. For each period of rotation, we first randomly determined the direction of rotation (clockwise or counterclockwise) and then randomly determined the rotation angle (between 0° and ±90°). For simplicity, we assumed that the fruit fly rotates with a constant velocity at $58.5°\,s^{-1}$, which corresponds to the aforementioned unilateral PB input of 3150 Hz with a maximum synaptic conductance of 0.3 nS. A constant rotational velocity allowed us to convert the rotation angle into the duration of the unilateral input.

**Calculation of the firing rate**. The firing rate $r(t)$ of a neural pool was calculated by performing a convolution of the spike function $\delta$ with an exponential kernel:

$$r(t) = \frac{1}{N} \int_0^\infty \sum_{i=0}^n \delta(t - \tau - t^i) e^{-\tau/d} \, d\tau,$$

where $N$ is the number of neurons in the pool, $t^i$ is the time of the $i$-th spike, and $n$ is the total number of spikes occurring in the neural pool. The decay time of the exponential kernel is defined by $d$. By using an exponential kernel with $d = 721.5$ ms, which indicates a 500 ms half-life, we intended to mimic the calcium signal from the popular calcium indicator GCaMP6f[70], which has a half-life of several hundred milliseconds. Such methods allowed us to compare the calculated population firing rate with the calcium image data reported in various studies, including that in Seelig and Jayaraman[10]. When plotting the activity of EB regions (in Figs. 3–5), the activity is represented by the mean firing rate of EIP neurons projecting to the given region.

**Code availability**. The code used for the model simulation is available from the corresponding author upon request.

**Data availability**. The data used to construct the proposed model are available within the paper and its Supplementary information files.

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

## Acknowledgements

We thank Dr Ann-Shyn Chiang for helpful discussions. We also thank the National Center for High-Performance Computing for providing computational resources. The work was supported by the Ministry of Science and Technology grants 105-2633-B-007-001 and 105-2311-B-007-012-MY3 and by the Aim for the Top University Project of the Ministry of Education.

## Author contributions

T.-S.S. performed simulations, analyzed the data, and prepared figures. W.-J.L. analyzed the data and prepared figures. Y.-C.H. developed the neural network simulator. C.-T.W. provided estimation of some model parameters. C.-C.L. designed the study and wrote the manuscript.

## Additional information

**Competing interests:** The authors declare no competing financial interests.

