## [Peer Review File · Nature Communications]

Reviewers' Comments:

Reviewer #1 (Remarks to the Author):

A Summary

Authors construct a model of path integration from recent neuroanatomical and imaging data available on *Drosophila*. They use existent sets of neurons which connect the ellipsoid body (EB) of the central complex with the protocerebral bridge and vice versa. According to the model a subset of neurons (C-ring) is responsible for maintaining an activity bump representing azimuth of the most salient landmark even if this landmark disappears from sight. Such a bump has been observed earlier in the ellipsoid body by calcium imaging. The other subset of neurons (P-ring) is responsible for shifting the activity bump in the EB against the direction of body rotation. The hypothesized model has been tested by constructing a computational model based on spiking neurons. It can reproduce the measurements by Seelig and Jayaraman based on calcium imaging in *Drosophila* (representation of the most salient landmark only, representation of rotational motion parameters, width of the activity peak, etc.). Parameters like the presentation duration, or the brightness of the landmark did not affect the representation in the EB. Testable predictions have been deduced.

B Originality

Overall, there is sufficient novelty in the model and the topic of path integration is hot in vertebrates and insects.

C Validity, Quality

The paper is clearly written and the results of the simulation are plausible. However, the paper would gain comprehensiveness by a clearer differentiation of where synaptic interactions occur (EB/PB) both in the text and in the figures (shape or color of arrows). Moreover, the circuitry of the so-called full model including three hypothesized ring neurons needs to be shown in the main manuscript. It is not even clear whether 1 or 16 per each of three ring neurons are needed and whether known features (a map-like characteristics) are necessary for the model. The discussion would gain from considering the influence of straight path lengths rather than discussing two more dimensions.

D n/a

E Conclusions

The model works fine and produces biologically plausible data.

F Suggested Improvements

Major

EB C-ring and P-ring: Is it possible to identify these new terms with the known R1 to R4 units?

Line 112-114: "...a connection from neuron A of the EIP class to B of the PEI class is always accompanied by a reverse connection from B to A (Fig.2C)." Reading the sentence I expected like if 2green is connected to 1red (red arrow) I will see also a connection back (double arrow) but the so-called symmetrical feedback seems to be 10red to 2green (black arrow)? Or do you mean they are reversely connected in the PB? What actually is the functional relation of two PEIs like 1red and 10red in the EB that are encircled? What does an arrow from green to red mean when green is postsynaptic and red is presynaptic in the EB – a connection in the PB? If so please add in line 718 "to a PEI or PEN neuron" in the PB.

Figs. 2C,D: The asymmetry in the "southern" EB segment (4 green numbers instead of two elsewhere) is not addressed in the text. Is this found in the anatomy of the EB?

Line 133 "EIP5 [arrow to] PEN5 [arrow to] EIP6 [arrow to] PEN6 [arrow to] EIP7" I can understand this only if the first and third connection is in the PB and the second and fourth connection in the EB. If this interpretation is correct, a specification of the arrows into EB and PB would be very helpful. Similar considerations apply to lines 135 to 137.

Can the model reproduce the relative positioning between a landmark in the outside world and its representation on the EB as observed by Seelig and Jayaraman? For each new experiment it is reported that the absolute position on the EB changed.

Line 222 "R[PEI] and R[PEN]": ring neuron groups have already names R1 to R4 and R4m or R4d. Please use preexisting terms, at least give concordance. They come in groups of around 16 to 18, and two ring neuron groups are shown to be

sensitive to the location of the landmark (map in the bulb; Seelig and Jayaraman, 2013), whereas your assumed function could be served by just three single ring neurons? Please discuss. If you need 16 ring neurons of each type and their position sensitivity, you will have to show the assumed circuitry.

Line 301-307: More plausible explanations are: (1) Inertia is sensed by halteres. (2) Internal signals called corollary discharge or efference copy for the body rotation are being used.

Line 320 and following: Data are from walking flies. Please discuss, that there may be a massive difference in the functions of the central complex for flight or walking. E.g. noduli activity showed up during flight but not during walking in an older 3H-deoxyglucose study (J COMP NEUROL (1994) 340: 255-268).

Discussion of how to deal with the other two dimensions seems rather academic and the arguments straight forward; moreover, the Seelig and Jayaraman data underlying this study are taken from walking flies. It would be more rewarding to discuss how authors will take the lengths of straight path increments between turns into account. This seems important to orient proper in a landscape.

Minor

Fig. 1C: For clarity, I would write “cue onset” also in lines “task 2” and “3”.

Is Fig. 1D needed at all? You could state that the neurons of Figs. 2A and 2B coexist in the EB.

Line 89: noduli (NO).

Line 679 circular level. (no hyphen)

Line 711: ...by green, red and blue lines, respectively.

Figs. 2C,D legend, line 717-718: “... red, blue, or brown arrows for EIP to PEI or EIP to PEN.” Three colors for two different connections. Why?

Line 146: computational model

Line 193: one side of the PB.

Fig. 4D y-axis: Rotational Speed

Line 221: one another

Fig.5C y-axis: rotation (rad) – space

Fig.5D right y-axis: rate (Hz) - space

Line 837: R[EIP] lesion results in

Line 352: LT (lateral triange, now bulb)

Line 360: drivers [32], and (space after comma)

G References

The model seems to have some structural similarity with a previously published model by “Haferlach, T., Wessnitzer, J., Mangan, M. & Webb, B. Evolving a neural model of insect path integration. *Adaptive Behavior* 15, 273–287 (2007)”. You may want to give credit to this earlier work.

H Clarity and Context

Abstract, introduction and conclusions seem appropriate to me.

Reviewer #2 (Remarks to the Author):

The manuscript by Su et al. investigates the neural circuit mechanism underlying angular patch integration in the ellipsoid body (EB) of *Drosophila Melanogaster* by building an anatomically realistic spiking neural network model of protocerebral bridge (PB)-EB system based on connectome data. Analysis of the connectome data revealed that the EB circuit consisted of coupled symmetric (C ring) and asymmetric (P ring) ring. Most importantly, the detailed PB-EB network model could successfully simulate the activity bumps during angular path integration as observed in vivo where the authors show that the symmetric feedback network was required for persistent activity bump on the visual cue while the asymmetric feedback network was required for keeping track of spatial orientation in the dark with self-motion cue. This study is novel in that it is the first computational study of the kind to fully model the EB and relate the neural network structure to the functional aspects of the fruit fly's cognitive behavior. However, there are a few points of careful consideration that may help improve the model and the manuscript.

1. Although the authors aimed to dissect the roles of C ring and P ring in angular patch integration, it seems that the authors have constructed their model to function as their prediction with a pre-fixed answer. It will be not only be more interesting but also more convincing if the authors conducted their simulation and presented their results by systematically “investigating” the structural differences in behavior. For example, in line 115-116 and in line 721-723 for Fig. 2D, the authors already attribute the feedback circuit of P ring to the “shifting activity bump” even before describing their results in Figs. 4. Similarly, in lines 115-116, they already assume that ring C would be capable of supporting persistent activity before presenting the simulation results in Fig. 3. Moreover, the authors test the effect of visual cues only on C ring and not on P ring, again, showing that the authors already have a strong premise that C ring is only for persistent activity bump. In fact, P ring can also show persistent activity bump in response to visual cue (Fig. 4A) in addition to bump shifting in response to body rotation in the dark (Fig. 4C). It seems that the asymmetric circuit of the P ring can perform the functions of C ring thus it is not entirely clear whether C ring’s persistent activity bump has a distinctive and differential functional role over P ring. Hence, it will be important to re-structure the logic development in writing the manuscript and also directly demonstrate whether results in Fig. 3A-C can be or cannot be replicated by P ring.

2. In Fig. 3C, the authors demonstrate that the activity bump in response to the moving visual cue across the visual field is tracked accordingly. In Fig. 3C, it appears that the movement of the visual cue response from R8 to R7 took ~ 2 seconds, (visual cue at R8 at 0 seconds at R7 at ~2 seconds). Given the fact that activity bump persists for at least 10 s when the visual cue is only presented for 1 s (cue off at 1 s) as shown in Fig. 3A, the activity bump in R8 should be persistently active in local feedback network of PB-EB-PB for a visual cue presented for ~2s. Thus, the moving visual cue in Fig. 3C should be expected to show moving persistent activity lasting for at least 10 seconds (depending on the speed at which the visual cue moves) since moving visual cue is essentially an instantaneous visual cue off. It would help the readers if the authors clearly state how fast the cue was moved and re-examine how the speed of the visual cue movement affects the tracking of the activity bump.

3. In Fig. 5, the authors connected C ring and P ring using GABAergic connections to construct a full EB-PB circuit model based on previous experimental immunocytochemistry. Here, the authors assume that GABAergic connections provide “global inhibition” (lines 403-404), presumably by GABAA receptor-mediated inhibition but provide no evidence or reference for this. Along the same line, NMDA, GABAA and acetylcholine receptors were included in the model but there is no justification for including acetylcholine in their model at all.

Moreover, it is difficult to follow how the parameters used in the model for the neurons and the synapses were determined and whether they are within the physiological range observed in the fruit fly *in vivo* and *in vitro*.

4. NMDA receptors were used as the main excitatory synapse in the model to show its relation to short-term memory of the visual cue. However, are NMDA receptor kinetics and its $[Mg^{2+}]$ -dependent gating mechanism essential for the activity bump phenomenon? Can other voltage-gated receptors such as AMPA receptor with appropriate parameters replace the role of NMDA receptors in this model? It would be useful to know how the NMDA receptor kinetics and its synaptic weight change influences on the “short-term memory”.

5. In Fig. 5, it is hard to compare the network function of C ring circuit, P ring circuit, and full EB-PB circuit. If the simulation paradigm in Fig. 3 and 4 were repeated in the full EB-PB circuit, it would be easier to show the full EB-PB circuit functions as a whole. Moreover, the role of GABAergic ring neurons also needs to be more clearly justified in full EB-PB circuit. In the current setting, the suppression of REIP neurons disrupts the persistent activity bump, which seems to totally overwrite the whole previous results in Fig. 3-4. Rather than starting with the full EB-PB circuit, it would be better if the authors introduce the GABAergic connections (REIP, RPEN, RPEI) one at a time to dissect the roles of how each GABAergic neuron contribute to the spatial orientation memory task. Then, the conclusion would become more convincing, providing the novel insight and predictions to not only the roles of excitatory but also inhibitory circuitry in EB.

6. In the central complex of the fruit fly, fan-shaped body (FB) gates the input from PB to EB. In addition, FB is known to be required for visual memory (Pan et al. 2009). It would be beyond the scope of this study to examine the effect of FB in this manuscript, but it would be good to include a paragraph discussing why FB was neglected in the present model and how the disynaptic excitatory pathway (PB-FB-EB) might affect the overall outcome. Similarly, justification or discussion on why other ring structures in the EB such as A, O rings were neglected would be nice.

Minor points:

1. This manuscript would benefit a lot if certain concepts and ideas were more explicitly and clearly explained with sufficient evidence and reasons. For examples, in introduction, line 41, it would help the readers to understand the concept of “activity bump” more clearly if the phenomenon was explained with an illustrative figure. Moreover, in the legend of Fig. 1A-C (line 665-658), in no where it states that the “activity bump” is represented as “yellow bars” in the visual field presented in “black”, for example. Other figures are not clearly explained as to what each figure component represents in the figure legends throughout the manuscript and the readers have to make sense by themselves.

2. In the stimulation paradigm, the authors inject the bilateral input to PB neurons as visual cue, but how this is biologically plausible input is not clearly stated.

3. Some typos:

Line 6: EP-PB → EB-PB

Line 221: on → to

Line 825: special → spatial

References:

Pan. Y., Zhou. Y., Guo. C., Gong. H., Gong. Z. & Liu. L. Differential roles of the fan-shaped body and the ellipsoid body in *Drosophila* visual pattern memory. *Learning & Memory*. 16(5):289-95 (2009)

Reviewer #3 (Remarks to the Author):

This paper presents an interesting model for how the central complex could maintain and update orientation information, based on the recent reports of this phenomenon in *Drosophila* by Seelig & Jayaraman. The concept draws also on recently published details of neural types and connectivity in this system by Wolf et al. As such, I believe the authors are not the only group to have arrived at this functional hypothesis (having seen several presentations at a recent conference that mention the same principle), but as far as I can see it has not yet been published by anyone, and there is no reason to doubt that these authors could have discovered it independently. It is certainly nice to see the principle confirmed in a computational model, and it is a very timely contribution.

I have a few major issues with the paper in its current form:

1) Although strong emphasis is placed on the model being based "upon detailed connectome analysis down to the single neuron level" the actual function depends on some rather more hypothetical connectivity assumptions (line 409 onwards) in particular the role of the ring neurons to switch between functions, including rather crucial anticipatory timing (lines 468-474) which is only justified by vague reference to primate eye saccade circuitry. It is also not clear why (line 436-7) each neuron type is simulated using a pool of 10 neurons. Are these not single neurons

in the anatomy, and if not, is the number 10 based on any specific data? Is it actually crucial to the circuit function to use a pool of 10 rather than single neurons? What other details of the neural model, e.g. inclusion of NMDA-dependent currents, are crucial, and why?

2) There is similar emphasis on having reproduced the empirical findings of Seelig & Jayaraman, but one striking feature of that data is not reproduced: the 'activity bump' in the fly is not retinotopically fixed, i.e., does not arise in a specific compartment mapping to 360 degree visual space, but can arise in any compartment (sometimes differently in the same fly) before tracking with the visual or motor input. In the model, the mapping is fixed (lines 449-450). This difference should be discussed, particularly as most of the other empirical effects described, such of drift in the position of the bump relative to the correct orientation over time (lines 233-234), are not all that unexpected (the drift simply reflects the integration of error over time).

Some minor points:

- The first sentence (lines 34-35) does not strike me as obviously true. If the animal can see the landmark, or regain sight of the landmark often enough, or associate another cue (e.g. sun direction) with its orientation to the landmark, it might not need to keep track of its own spatial orientation to successfully get to the goal.

- The suggestion that the input for shifting the bump through self-movement is from the JO (lines 301-313) is interesting, although it seems incompatible with the requirement for anticipatory movement information mentioned above (lines 468-473). It also seems quite crucial that this input is nicely tuned - in practice this is hand-tuned in the model (lines 486-488) but how might it be tuned by the fly?

- line 221, 'on' should be 'one'; line 338 'neuron' should be 'neuron'

We are glad that the reviewers are positive to our study and we thank them for providing constructive comments that greatly help us to improve the manuscript. We have substantially revised the manuscript with all changes highlighted (in red). We have revised Figures 1-5 by adding or replacing 11 panels and we have also added more than 100 lines of text to the manuscript. Please see our point by point response below.

Reviewer #1

C. Validity, Quality

1. *The paper is clearly written and the results of the simulation are plausible. However, the paper would gain comprehensiveness by a clearer differentiation of where synaptic interactions occur (EB/PB) both in the text and in the figures (shape or color of arrows). Moreover, the circuitry of the so-called full model including three hypothesized ring neurons needs to be shown in the main manuscript.*

Reply: We thank the reviewer for the suggestion. We replot Figures 2C and 2D by clearly indicating where the synaptic interactions occur and also by including the ring neurons in the Figure 2A and 2B. Note that Figure 2A and 2B are reconstruction of anatomical data and therefore we only plot the known ring neuron types (R1 and P-ring). In the model we hypothesize that the R1 and P-ring ring neurons can be functionally categorized into R_EIP, R_PEI and R_PEN types. We have also clearly stated the hypothesis in lines 91-95, 497-500.

2. *It is not even clear whether 1 or 16 per each of three ring neurons are needed and whether known features (a map-like characteristics) are necessary for the model.*

Reply: The short answers to both questions are no. But please see our reply to the comment #9 below for a detailed discussion on these issues.

3. *The discussion would gain from considering the influence of straight path lengths rather than discussing two more dimensions.*

Reply: We thank the reviewer for the great suggestion. We discuss this issue in detail in our response to the comment #12 below.

F. Suggested Improvements

Major:

4. *EB C-ring and P-ring: Is it possible to identify these new terms with the known R1 to R4 units?*

Reply: In Fig. S1 of “Lin, C.-Y. et al. Cell Rep. 3, 1739–1753 (2013),” the authors compared their definition of EB rings and that in Renn et al, 1999. According to this figure, the C ring corresponds to R1 defined in Renn et al, 1999, while the P ring corresponds to an undefined unit in Renn et al. 1999. We add this information in lines 491-493.

5. *Line 112-114: “...a connection from neuron A of the EIP class to B of the PEI class is always accompanied by a reverse connection from B to A (Fig.2C).” Reading the sentence I expected like if 2green is connected to 1red (red arrow) I will see also a connection back (double arrow) but the so-called symmetrical feedback seems to be 10red to 2green (black arrow)? Or do you mean they are reversely connected in the PB? What actually is the functional relation of two PEIs like 1red and 10red in the EB that are encircled? What does an arrow from green to red mean when green is postsynaptic and red is presynaptic in the EB – a connection in the PB? If so please add in line 718 “to a PEI or PEN neuron” in the PB.*

Reply: We are sorry for the confusion. We completely replot figure 2C (and 2D) with improved representation to clearly indicate the synaptic connections between each type of neurons and where the connections locate. As the reviewer expected, the 2green → 1red connection is indeed accompanied by a reverse connection, 1red → 2green, which is indicated by the black arrow in the original plot. We believe that it is much easier for the readers to understand the detailed circuit organization in the new plot.

6. *Figs. 2C,D: The asymmetry in the “southern” EB segment (4 green numbers instead of two elsewhere) is not addressed in the text. Is this found in the anatomy of the EB?*

Reply: Yes, the “non-typical” connections formed by the two extra neurons, EIP0 and EIP17, in the southern EB segment are observed and reported in Wolf et al. J. Comp. Neurol. 523, 997–1037 (2015). The existence of the two neurons does not affect the dynamics of the EB-PB system we reported here and their exact function remains unclear. We add the above information in lines 413-416.

7. *Line 133 “EIP5 [arrow to] PEN5 [arrow to] EIP6 [arrow to] PEN6 [arrow to] EIP7” I can understand this only if the first and third connection is in the PB and the second and fourth connection in the EB. If this interpretation is correct, a specification of the arrows into EB and PB would be very helpful. Similar considerations apply to lines 135 to 137.*

Reply: The interpretation is correct and we thank the reviewer for the suggestion. We add a label to each arrow to specify the brain region (now in lines 138-139 and 141-142).

8. *Can the model reproduce the relative positioning between a landmark in the outside world and its representation on the EB as observed by Seelig and Jayaraman? For each new experiment it is reported that the absolute position on the EB changed.*

Reply: In the current model we do not consider the variation of the offset (between the bump position in EB and the orientation of the landmark) across trials and across individuals observed in Seelig and Jayaraman 2015. The observation suggests that the EB does not have a fixed retinotopic mapping of the surroundings. We hypothesize that the variable mapping is not the result of neural computation in the EB-PB circuits but rather due to an upstream circuit which flexibly remaps the retinotopic input to fruit fly's internal representation of the space after each trial or after certain environmental change. We plan to address this issue in our follow-up study which will expand the model circuits to the surrounding neuropils. However, we would like to stress that the variability of the bump-landmark offset does not affect the core model feature: angular integration by shifting the EB bump at any location under the unilateral input to PB. We have added this argument in Discussion (lines 347-353).

9. *Line 222 “R[PEI] and R[PEN]”: ring neuron groups have already names R1 to R4 and R4m or R4d. Please use preexisting terms, at least give concordance. They come in groups of around 16 to 18, and two ring neuron groups are shown to be sensitive to the location of the landmark (map in the bulb; Seelig and Jayaraman, 2013), whereas your assumed function could be served by just three single ring neurons? Please discuss. If you need 16 ring neurons of each type and their position sensitivity, you will have to show the assumed circuitry.*

Reply: In the current model we only assumed three ring neuron types and each type consists of a pool of 10 identical neurons. We did not simulate ring neurons' selectivity of landmark features (location, orientation, etc) as observed in Seelig and Jayaraman 2013 because 1) in the present study our goal is to reproduce the EB bump dynamics in the simplest environmental setting (with only one or two simple visual cues), and 2) the selectivity was observed for R2 and R3/R4 ring neurons, while in the model the three ring neuron types are assumed to be subsets of R1 and P-ring ring neurons. Therefore, at the current stage we cannot make the correlation between the modeled ring neurons and the observed visual feature selectivity. However, we hypothesize that the R2 and R3/R4 ring neurons modulate neural activity in EB when a fly moves in a space with complex landmark features. We do have a plan to include the selectivity of ring neurons in a model extension in a follow-up study. We have revised the discussion about ring neuron selectivity and include the information above in lines 497-500 and 360-366.

Another question is whether we need exactly 10 neurons in each ring neuron type. The short answer is no. From the neural computation point of view, we just need several neurons that deliver a total firing rate large enough to sustained excitation or inhibition for the downstream neurons. Ring neurons form GABAergic synapses, which have an activation time constant about 5ms. Therefore we need at least 200 spikes per second in order to maintain stable inhibition. Considering that central complex neurons in other insets are found to have an average firing rate about several tens of spikes per second (for example: Sakura et al. J Neurophysiol 99,667-682 (2008); Homberg et al Phil Trans R Soc B 366, 680-687 (2011)), 10 neurons per neuron type will provide a sufficient total spike rate to the downstream neurons. It is also reasonable to assume that the brain maintain several neurons per neuron type as redundancy, otherwise any damage to a neuron could cause significant impact to the related brain function. We have added the argument above in lines 526-534.

Regarding the naming of the ring neurons in the model, R_{PEI} innervates the C ring and is therefore corresponds to the R1 neurons. However, R_{PEN} innervates the P ring, which does not correspond to any of the R1-R4 units but the P ring innervating neurons have been observed both in Lin, C.-Y. et al. 2013 and in Wolff & Rubin 2015. R2 and R3 neurons innervate the A ring whereas R4d and R4m neurons innervate the O ring. Both A ring and O ring neurons are not modeled in the current model. We have provide such information in lines 497-499 and 497-

500.

10. *Line 301-307: More plausible explanations are: (1) Inertia is sensed by halteres. (2) Internal signals called corollary discharge or efference copy for the body rotation are being used.*

Reply: We thank the reviewer for the valuable comment. Indeed, JO may not be a good source of the unilateral input to PB during the body rotation in the dark. We agree that the efference copy of the motor commands from the downstream motor regions and the halteres are likely the better sources for the PB unilateral input. We have rewritten the related text accordingly (lines 324-334).

11. *Line 320 and following: Data are from walking flies. Please discuss, that there may be a massive difference in the functions of the central complex for flight or walking. E.g. noduli activity showed up during flight but not during walking in an older 3H-deoxyglucose study (J COMP NEUROL (1994) 340: 255-268).*

Reply: Thanks for the suggestion. Behavior-dependent sensory response is indeed an important issue. We reviewed related papers from Heisenberg lab (Bausenwein et al. *J. Comp. Neurol.* 340, 255–268 (1994)) and Dickinson lab (Weir et al. *J. Neurophysiol.* 111, 62–71 (2014); Weir and Dickinson et al. *Proc. Natl. Acad. Sci. U. S. A.* 112, E5523–E5532 (2015)) and found that the existing observations are in line with our working hypothesis, in which only a subset of EB ring neurons are required for walking on a horizontal plane with simple stimuli. When we consider flight in a 3D environment, orientation can be changed by manipulating yaw, pitch and roll. To keep track of orientation, a fly requires computation from more EB rings and/or more surrounding regions such as the fan-shaped body and noduli. We strengthen the related discussion in the manuscript (lines 367-376) by including the references and argument above.

12. *Discussion of how to deal with the other two dimensions seems rather academic and the arguments straight forward; moreover, the Seelig and Jayaraman data underlying this study are taken from walking flies. It would be more rewarding to discuss how authors will take the lengths of straight path increments between turns into account. This seems important to orient properly in a landscape.*

Reply: We totally agree with the reviewer. It is important to consider how a straight path is integrated and how it is combined with angular integration to

provide signals for the navigation system in the brain. Unfortunately, there is still not enough data on straight path integration in *Drosophila* and therefore it is too early for us to construct a data-driven model. However, data presented in Weir and Dickinson et al. Proc. Natl. Acad. Sci. U. S. A. 112, E5523–E5532 (2015) provide some hints on this aspect. Their study shows that FB exhibits strong activity during flight and subsets of FB neurons response to progressive and regressive optical flows. Such neural responses may provide signals for the straight path integration. On the other hand, the efference copy of forward or backward movement command should also play an important role, in particular when a fly moves in the dark. More anatomical and functional data are required in order to identify the circuits that are responsible for the straight path integration. We hypothesize that the FB play roles in such integration and we are currently conducting a connectomic study on related circuits. We are also examining the circuits between EB and other neuropils such as lateral triangles. The neural mechanisms about path integration and the “cognitive map” in rodents discussed in McNaughton et al. Nat Rev Neurosci 7, 663-678 (2006) provide some insights into this issue. However, we would like to emphasize that the network models of path integration and navigation for rodents were proposed without the availability of cellular level connectome. The studies in *Drosophila* provide a unique opportunity to verify whether the proposed circuits exist in a much simpler brain. We add the discussion above in lines 377-389.

Minor:

Fig. 1C: For clarity, I would write “cue onset” also in lines “task 2” and “3”. Is Fig. 1D needed at all? You could state that the neurons of Figs. 2A and 2B coexist in the EB.

Line 89: noduli (NO).

Line 679 circular level. (no hyphen)

Line 711: ...by green, red and blue lines, respectively.

Figs. 2C,D legend, line 717-718: “... red, blue, or brown arrows for EIP to PEI or EIP to PEN.” Three colors for two different connections. Why?

Line 146: computational model Line 193: one side of the PB. Fig. 4D y-axis: Rotational Speed Line 221: one another

Fig.5C y-axis: rotation (rad) – space Fig.5D right y-axis: rate (Hz) - space Line 837:

R[EIP] lesion results in Line 352: LT (lateral triangle, now bulb) Line 360: drivers [32], and (space after comma)

Reply: We thank the reviewer for pointing out these typos and errors, and we have addressed all of them. We have removed Fig. 1D and replaced it with a schematic that illustrates the basic interaction between different classes of neurons in the circuit. Figs 2C & 2D have been replotted and the legends have been rewritten.

G. References

The model seems to have some structural similarity with a previously published model by “Haferlach, T., Wessnitzer, J., Mangan, M. & Webb, B. Evolving a neural model of insect path integration. Adaptive Behavior 15, 273–287 (2007)”. You may want to give credit to this earlier work.

Reply: We thank the reviewer for providing the reference. We have cited it at lines 64 and 387. This work focused on how the information about heading can be recorded in a neural circuit when the visual information about direction is available and how this memory helps an agent return to its nest. Our model focuses on how heading information can be recorded with the absence of visual information but the model does not address the homing behavior. Although with different focus, the two models both implement ring-like neural circuits and rely on the asymmetric neural activity to update the system. In Haferlach et al 2007, asymmetry is used to change the heading of the agent when homing while in our model, asymmetry is used to move the activity bump.

Reviewer #2

Major points:

1. **a.** *Although the authors aimed to dissect the roles of C ring and P ring in angular patch integration, it seems that the authors have constructed their model to function as their prediction with a pre-fixed answer. It will be not only be more interesting but also more convincing if the authors conducted their simulation and presented their results by systematically “investigating” the structural differences in behavior. For example, in line 115-116 and in line 721-723 for Fig. 2D, the authors already attribute the feedback circuit of P ring to the “shifting activity bump” even before describing their results in Figs. 4. Similarly, in lines 115-116, they already assume that ring C would be capable of supporting persistent activity before presenting the simulation results in Fig. 3. Moreover, the authors test the effect of visual cues only on C ring and not on P ring, again, showing that*

the authors already have a strong premise that C ring is only for persistent activity bump. b. In fact, P ring can also show persistent activity bump in response to visual cue (Fig. 4A) in addition to bump shifting in response to body rotation in the dark (Fig. 4C). It seems that the asymmetric circuit of the P ring can perform the functions of C ring thus it is not entirely clear whether C ring's persistent activity bump has a distinctive and differential functional role over P ring. Hence, it will be important to re-structure the logic development in writing the manuscript and also directly demonstrate whether results in Fig. 3A-C can be or cannot be replicated by P ring.

Reply: This is a good point and we thank the reviewer for the suggestion. In fact, we did perform systematic tests over both rings to verify their functional differences during the development of the model. We found that the C ring bump did not move when we applied unilateral excitation to one side of PB (shown in new Fig 3F). Although the P ring did exhibit persistent bump activity like the C ring did, we found that the bump was wider (less accurate), unstable and occasionally drifted (shown in new Figs 4A, 4B, 4D and 4E). We did briefly mentioned the instability of the P ring but now we expand the text (lines 203-209) to elaborate why P ring alone is not enough. We also re-structure the logic of the paper. The functional difference between the C and P rings is now presented as an assumption (not pre-fixed answer) proposed based on anatomical data (symmetric vs. asymmetric feedback connections)(lines 153-155). The assumption is then systematically tested for both rings as shown in the replotted Figs 3 & 4.

- 2. In Fig. 3C, the authors demonstrate that the activity bump in response to the moving visual cue across the visual field is tracked accordingly. In Fig. 3C, it appears that the movement of the visual cue response from R8 to R7 took ~ 2 seconds, (visual cue at R8 at 0 seconds at R7 at ~2 seconds). Given the fact that activity bump persists for at least 10 s when the visual cue is only presented for 1 s (cue off at 1 s) as shown in Fig. 3A, the activity bump in R8 should be persistently active in local feedback network of PB-EB-PB for a visual cue presented for ~2s. Thus, the moving visual cue in Fig. 3C should be expected to show moving persistent activity lasting for at least 10 seconds (depending on the speed at which the visual cue moves) since moving visual cue is essentially an instantaneous visual cue off. It would help the readers if the authors clearly state how fast the cue was moved and re-examine how the speed of the visual cue*

movement affects the tracking of the activity bump.

Reply: The C ring circuit can maintain a persistent activity bump after cue offset but can also quickly update the bump position when the cue is moving. This seemingly contradictory behavior is in fact the characteristics of the classic “attractor” ring network, which is exactly how the C ring circuit organized based on the anatomical data. In this type of networks, the local recurrent (feedback) excitation and global inhibition help to maintain a localized bump activity under a static stimulus. When the visual stimulus moves to an adjacent position, a new activity bump forms through the recurrent excitation at the new location and the old bump is quickly suppressed by the global inhibition. Of course, the cue cannot move too fast otherwise it does not stay in a location long enough for a new activity bump to form and the old bump to be suppressed. In Figure 3C the cue moved at $11.25^\circ/\text{s}$, which corresponds to 0.5 EB region per second. We have performed new simulations and found that the C ring circuit can track a moving cue up to $22.5^\circ/\text{s}$, or 1 EB region per second. We have revised the text in lines 172-179 to explain why the ring circuit can track a moving cue and also mention the speed limit.

3. **a.** *In Fig. 5, the authors connected C ring and P ring using GABAergic connections to construct a full EB-PB circuit model based on previous experimental immunocytochemistry. Here, the authors assume that GABAergic connections provide “global inhibition” (lines 403-404), presumably by GABAA receptor-mediated inhibition but provide no evidence or reference for this. Along the same line, NMDA, GABAA and acetylcholine receptors were included in the model but there is no justification for including acetylcholine in their model at all.*
- b.** *Moreover, it is difficult to follow how the parameters used in the model for the neurons and the synapses were determined and whether they are within the physiological range observed in the fruit fly in vivo and in vitro.*

Reply: We thank the reviewer for the comment. In the model, only the ring neurons are assumed to be GABAergic. The assumption is based on previous studies (Hanesch et al. Cell Tissue Res. 257, 343-366 (1989)). Moreover, each ring neuron infiltrates all regions in the ring it innervates. Therefore it is reasonable to assume that ring neurons provide global inhibition by making contacts with all other neurons which have dendritic innervation in the same ring. The expression of vesicular glutamate transporters, an indication of glutamatergic neurons, are found in various regions in the central complex including EB and PB (Kahsai and

Winther J. Comp. Neurol. 519, 290-315 (2011); Daniels et al. J. Comp. Neurol. 508, 131-152 (2008)). Moreover, acetylcholine markers are also expressed in EB and PB (Kahsai and Winther J. Comp. Neurol. 519, 290-315 (2011); Lin et al. Cell Rep. 3, 1739-1753 (2013)). We have revised the text and added the related references to the manuscript in lines 422-432 and 487-491.

Regarding the model parameters, they can be classified into three categories: 1) Neuron membrane related parameters. There was no direct measurement of these parameters in the central complex neurons. But we used typical values that are consistent with data for neurons in the fruit fly olfactory system. We stress that the exact values of these parameters are not crucially important in the neural network models because our goal is not to reproduce accurate electrophysiological properties of individual neurons but rather to demonstrate the behavior of the circuit, which could work in a large range of parameter values. We added text to explain how the parameters were determined in lines 448-456. 2) Synaptic time constants and reversal potential are all in commonly accepted ranges that are widely used in computational neuroscience studies. These ranges are pretty much preserved across species. 3) The synaptic strengths are tuning parameters. The synaptic strengths vary a lot across neurons and the values are not available for neurons in the central complex. In fact, due to the difficulty in measuring synaptic strength, it is not available for the majority of the neurons in any species. Therefore it is a common practice to treat the synaptic strengths as tuning parameters and they are tuned until the desired function of the model is achieved. The determination of the synaptic strength was explained in lines 509-523.

4. *NMDA receptors were used as the main excitatory synapse in the model to show its relation to short-term memory of the visual cue. However, are NMDA receptor kinetics and its [Mg²⁺]-dependent gating mechanism essential for the activity bump phenomenon? Can other voltage-gated receptors such as AMPA receptor with appropriate parameters replace the role of NMDA receptors in this model? It would be useful to know how the NMDA receptor kinetics and its synaptic weight change influences on the "short-term memory".*

Reply: According to the well-established theories for working/short-term memory and decision making (Wang, J Neurosci. 19:9587-9603 (1999); Wang, Neuron. 60:215-234 (2008)), a long time constant is required for sustaining the activity bump when the firing rate is limited to a physiological range (several tens of spikes per second). This is why many attractor models have to rely on the

NMDA receptors, which have a long time constant of ~ 100 ms (compared to merely 2ms for AMPA receptors). The $[Mg^{2+}]$ -dependent gating mechanism is, in fact, not essential to the bump attractor mechanism. Therefore, in theory, one can replace NMDA with other receptors with a long time constant. AMPA receptor's time constant is simply too short to work. Acetylcholine receptors, on the other hand, have been reported to exhibit a moderate time constant up to 20ms and therefore may serve as an alternative choice. But they need to sustain an activity bump with a much higher firing rate. We performed a simulation by replacing NMDA with AMPA receptors. We adjusted the synaptic weights so that the mean synaptic current of the AMPA-only model is the same with the original model. However, the bump could not be sustained and it quickly spread to the entire ring after the stimulus offset. The new result is included in Figure 5E and we describe the result in lines 271-282.

5. *In Fig. 5, it is hard to compare the network function of C ring circuit, P ring circuit, and full EB-PB circuit. If the simulation paradigm in Fig. 3 and 4 were repeated in the full EB-PB circuit, it would be easier to show the full EB-PB circuit functions as a whole. Moreover, the role of GABAergic ring neurons also needs to be more clearly justified in full EB-PB circuit. In the current setting, the suppression of REIP neurons disrupts the persistent activity bump, which seems to totally overwrite the whole previous results in Fig. 3-4. Rather than starting with the full EB-PB circuit, it would be better if the authors introduce the GABAergic connections (REIP, RPEN, RPEI) one at a time to dissect the roles of how each GABAergic neuron contribute to the spatial orientation memory task. Then, the conclusion would become more convincing, providing the novel insight and predictions to not only the roles of excitatory but also inhibitory circuitry in EB.*

Reply: We thank the reviewer for the suggestion. Indeed, it is important to clarify the role of each type of GABAergic ring neurons in spatial orientation memory. The R_{EIP} ring neurons are essential in sustaining the activity bump both in C and P rings because R_{EIP} neurons exhibit moderate global inhibition which helps focusing and stabilizing the activity bump. R_{EIP} is needed in C-ring circuit and P-ring circuit (and of course also in the full model) because EIP neurons are shared in both rings.

The other two ring neuron types have different functions and are only needed in the full EP-PB model. The purpose of the full model is to combine the advantages

of both C-ring and P-ring circuits for more realistic movement conditions which consist of alternate forward walking and turning. Therefore, in the full model C-ring and P-ring circuits operate alternatively: the C ring operates during forward walking when the orientation does not change and a stable activity bump is sustained. On the other hand, when a fly makes a turn, The P ring has to operate in order to update the orientation by shifting the activity bump. Such alternation is achieved by the two ring neuron types, R_{PEI} and R_{PEN} , which strongly inhibit PEI neurons (C ring) and PEN neurons (P ring), respectively. Therefore, these two types of ring neurons are only implemented in the full model and their function is to alternate the activity of P and C rings according to the movement states of the flies. We think the original Figure 5D serves perfectly the purpose of illustrating the role of each ring neuron type in spatial orientation memory. Therefore we moves Figure 5D forward to Figure 5A and now we start the full model with introducing the role of each ring neuron type. We also rewrote the related text (in lines 238-260) to provide a much better explanation on the functions of the ring neurons and on how the full model works as a whole. Due to the nature of alternate operation, the full model performs exactly like the C ring during forward walking (because the P ring is inhibited) but like the P ring when the body is rotating (because the C ring is inhibited). Therefore we do not repeat every test for the full model as we did for the C ring and P ring. However, Fig 5C already covers the most important test which shows how the full model performs after the stimulus offset and with body rotation.

6. *In the central complex of the fruit fly, fan-shaped body (FB) gates the input from PB to EB. In addition, FB is known to be required for visual memory (Pan et al. 2009). It would be beyond the scope of this study to examine the effect of FB in this manuscript, but it would be good to include a paragraph discussing why FB was neglected in the present model and how the disynaptic excitatory pathway (PB-FB-EB) might affect the overall outcome. Similarly, justification or discussion on why other ring structures in the EB such as A, O rings were neglected would be nice.*

Reply: We thank the reviewer for the suggestion. Based on several previous studies (Weir et al 2014; Weir and Dickinson et al. 2015), the FB responds to complex visual stimuli and are more activated during flight than during walk. Since in the present study we focus on the spatial orientation under simple

stimuli during walk, it is justify that we neglect FB for the moment. However, we recognize the importance in addressing spatial orientation under complex visual stimuli during flight, which involves more complex neural computation because the animals move in three dimensions. We hypothesize that when the flies move in the three dimensional space under a complex visual environment, other EB rings and the FB participate in the neural computation that keeps the flies oriented. We will test the hypothesis in an extension of our model in the follow up study. We touched this issue briefly in the original manuscript and now we provide more details in lines 354-376.

Minor points:

1. *This manuscript would benefit a lot if certain concepts and ideas were more explicitly and clearly explained with sufficient evidence and reasons. For examples, in introduction, line 41, it would help the readers to understand the concept of “activity bump” more clearly if the phenomenon was explained with an illustrative figure. Moreover, in the legend of Fig. 1A-C (line 665-658), in no where it states that the “activity bump” is represented as “yellow bars” in the visual field presented in “black”, for example. Other figures are not clearly explained as to what each figure component represents in the figure legends throughout the manuscript and the readers have to make sense by themselves.*

Reply: We fully agree that it would help the readers if we add an illustration for the activity bump concept. Such illustration is now included as Figure 1C. We have revised all figure legends to explain each figure component.

2. In the stimulation paradigm, the authors inject the bilateral input to PB neurons as visual cue, but how this is biologically plausible input is not clearly stated.

Reply: Although the visual input pathways to the central complex is not fully understood, there are four suggested pathways (Lin et al. Cell Reports 2013). Most of these pathways enter the central complex through large-field inhibitory (GABAergic) or modulatory (dopaminergic) neurons except for one excitatory pathway that enters PB local glomeruli by small-field cholinergic neurons. To elicit a localized bump activity, large-field inhibitory or modulatory neurons are not likely to work. Therefore we chose to implement the visual input via PB in light of this excitatory pathway. The input has to be injected into two bilateral FB regions

because they converge to the same EB region symmetrically. Otherwise, a unilateral input to PB will cause a moving activity bump even when the visual cue is not moving. Another indirect support comes from the observation that in locust the PB responds to the E-vector of the polarized light in a bilateral fashion, suggesting the visual input to PB is organized bilaterally (Heinze and Homberg. *Science* 315, 995-997 (2007); Homberg et al. *Phil. Trans. R. Soc. B.* 366, 680-687 (2011)). We would like to emphasize that from the modeling point of view, injecting the input either to two bilateral PB regions or to the corresponding EB region produces the same result (elicits the same activity bump). Therefore if a future study identify a new visual pathway to individual EB regions responsible for eliciting the activity bump, we only need to change our visual input scheme and the proposed circuit principle about spatial orientation memory will not be affected. We add the argument above in the discussion section (lines 550-565).

3. Some typos:

Line 6: EP-PB → EB-PB Line221: on → to

Line 825: special → spatial

Reply: Thanks for pointing out the typos. They have all been corrected.

References:

Pan. Y., Zhou. Y., Guo. C., Gong. H., Gong. Z. & Liu. L. Differential roles of the fan-shaped body and the ellipsoid body in Drosophila visual pattern memory. Learning & Memory. 16(5):289-95 (2009)

Reply: We have included this reference. It is cited in line 364.

Reviewer #3

Major points:

1. *Although strong emphasis is placed on the model being based "upon detailed connectome analysis down to the single neuron level" the actual function depends on some rather more hypothetical connectivity assumptions (line 409 onwards) in particular the role of the ring neurons to switch between functions, including rather crucial anticipatory timing (lines 468-474) which is only justified by vague reference to primate eye saccade circuitry. It is also not clear why (line*

436-7) each neuron type is simulated using a pool of 10 neurons. Are these not single neurons in the anatomy, and if not, is the number 10 based on any specific data? Is it actually crucial to the circuit function to use a pool of 10 rather than single neurons? What other details of the neural model, e.g. inclusion of NMDA-dependent currents, are crucial, and why?

Reply: We thank the reviewer for the comment. We emphasized “cellular-level connectome” because we would like to distinguish our work from the conventional neural network modeling. In the present study, we first had the cellular-level connectome of a brain region in hand, and then we infer its working principle by carefully analyzing the circuits based on several assumptions. In conventional neural network modeling, people first have a brain function in mind, and then they construct neural circuits to realize the function with very little or no cellular-level connectome available. But to address the reviewer’s comment, we toned down some statements and also clearly state the assumptions and hypotheses, and explain the rationale in Methods (lines 424-428, 449-453 and 478-485).

Regarding simulating each neuron type with a pool of 10 neurons, the model requires multiple neurons per type but the exact number is not essential. This requirement is based on several considerations: 1) it is a common observation that a postsynaptic action potential is only triggered by a barrage of spikes input from multiple presynaptic neuron spikes. If each neuron type in the central complex only consists of one neuron, it will not provide enough input to activate the downstream neuron given that the typical firing rate of a central complex neuron under visual stimuli is in the range of several tens of Hertz. 2) The variability of the spike timing of a single neuron is often large. By sampling spikes from a large number of upstream neurons rather than from just one, a neuron can produce a much reliable response to a sensory stimulus. 3) It is also make sense to have redundancy. If each neuron type consists of only one neuron, damage to a single neuron can easily break the ring circuits and impair the entire function of orientation memory. We have provided the argument above in lines 526-534.

Inclusion of NMDA receptors in the model is crucial in terms of the long time constant (~100ms) the receptors provide. It is a well-established theory that a long receptor time constant is required in order to maintain activity bumps in a recurrent neural network (Wang, *J Neurosci.* 19:9587-9603 (1999); Wang, *Neuron.* 60:215-234 (2008)). However, any other excitatory receptor that can

provide long time constant may also work. For example, some acetylcholine receptors are claimed to have a time constant up to 20 ms. We would like to emphasize that using either NMDA or the long time constant acetylcholine receptors does not alter the computational principle we proposed here. We demonstrate the importance of the NMDA receptors in lines 271-282 and in Fig 5E.

- 2. There is similar emphasis on having reproduced the empirical findings of Seelig & Jayaraman, but one striking feature of that data is not reproduced: the 'activity bump' in the fly is not retinotopically fixed, i.e., does not arise in a specific compartment mapping to 360 degree visual space, but can arise in any compartment (sometimes differently in the same fly) before tracking with the visual or motor input. In the model, the mapping is fixed (lines 449-450). This difference should be discussed, particularly a most of the other empirical effects described, such of drift in the position of the bump relative to the correct orientation over time (lines 233-234), are not all that unexpected (the drift simply reflects the integration of error over time).*

Reply: We totally agree with the reviewer. This is indeed an important issue and we plan to address this issue in our future study. We think that the variable mapping phenomenon involve very complex neural computation in the brain and deserves another research project when more data are available. We suspect that EB encodes the fly's sense about its orientation in the absolute space, or an internal representation of the space, which will change or reset after each trial or after some environmental change. There should be a circuit upstream to EB and/or PB, and flexibly remaps the retinotopical input to the fly's internal representation of the space. We have added the argument above in Discussion (lines 347-353).

Minor points:

- 1. The first sentence (lines 34-35) does not strike me as obviously true. If the animal can see the landmark, or regain sight of the landmark often enough, or associate another cue (e.g. sun direction) with its orientation to the landmark, it might not need to keep track of its own spatial orientation to successfully get to the goal.*

Reply: Indeed, it may not be necessary for an animal to keep track of its

orientation if the visual contact with the landmark is always available. So we changed the sentence to address the reviewer's concern. However, we believe in natural environment the visual contact with the landmark can often get lost. For example, when an insect flies from an open field into woods, the insect loses its visual contact with the sun and has to find another salient visual object for guidance. Without the ability of short-term/working memory, a neuron can lose its response to a visual stimulus within a 100 ms when the visual contact is temporarily lost. In fact, we believe that such a temporary loss of visual contact with the landmark and the transition from one landmark to another occur so frequently that an animal has to constantly maintain its sense about the orientation in order to smoothly move in natural environment.

2. *The suggestion that the input for shifting the bump through self-movement is from the JO (lines 301-313) is interesting, although it seems incompatible with the requirement for anticipatory movement information mentioned above (lines 468-473). It also seems quite crucial that this input is nicely tuned - in practice this is hand-tuned in the model (lines 486-488) but how might it be tuned by the fly?*

Reply: We totally agree with the reviewer. JO may not be a good (at least not the only one) source of the information about the body movement. As the reviewer #1 suggested, the efference copy of the motor commands may be a better source for the unilateral input to PB that shifts the bump. In fact, one of the major input to the PB is from DMP (dorsalmedial protocerebrum), which is known to be involved in motor functions. Several studies on primates have demonstrated that efference copy can elicit response in the upstream brain region before the occurrence of an actual motor action (Crapse and Sommer. *Nat. Rev. Neurosci.* 9, 587-600 (2008); Sommer and Wurtz. 296, 1480-1482 (2002)). The function may be to help the neural circuits pre-adapt to the change of sensory input due to the upcoming motor action. We have revised the Discussion in lines 324-333.

Regarding how this input can be tuned in fly, we believe that it has to be tuned during development. The central complex appears (or becomes identifiable) in the third instar larvae and matures after metamorphosis. If in the beginning of the development the synaptic strength is incorrect, it causes a mismatch between the visual cue movement and the movement of the activity bump. We hypothesize that some form of synaptic plasticity will take place and gradually

modify the synaptic strength until the two movements match. We added our argument in lines 341-346.

3. *line 221, 'on' should be 'one'; line 338 'neuron' should be 'neurons'*

Reply: The typos have been corrected. Thanks.

Reviewers' Comments:

Reviewer #1 (Remarks to the Author):

Overall, my points of critique have been adequately addressed in text and figures. I have only minor suggestions to some revised parts of the text.

l.66: "At the single-cell level..."

l.171: "...0.5 EB regions per ..."

l.174: "...as a series of cue-offset events each followed by a cue-onset event at an adjacent position."

l.175/176: "...while it suppresses..."

l.251: "...ring-neuron types..."

l.326: "...by the efference copy or a reafferent signal of the rotation..."

l.338: "...into angular velocity information."

l.346/347: "synaptic plasticity will take place" To me "synaptic plasticity" seems rather a property of a neuron than a process. "plastic changes in synaptic strength"?

l.349: "...bump in the EB..."

l.353: "...to the fruit fly's..."

l.363: "...in the EB..."

l.374: "...state-dependent responses..."

l.377: "...space is represented..."

l.391: "...cellular level."

l.417: "...such a-typical..." or "...non-typical..."

l.425: "...EB play a crucial role..."

l.427: "...that the mutual excitation..."

l.428: Not the markers are expressed; "constituents of the ACh machinery"?

l.432: Neurotransmitter-receptors rather than neurotransmitters determine whether a synapse is excitatory or inhibitory. Please rephrase.

l.531: "(3) it does make sense..."

l.558: "...that in locusts..." or "in the locust"

l.562: "...future study identifies..."

l.840: "...innervate the C ring..."

l.883: "...sub-circuit."

Fig.3G right axis labels need more space.

l.1002: "...two ring-neuron classes."

Reviewer #2 (Remarks to the Author):

The authors made a lot of effort to address the major concerns that were raised previously by adding clearer explanations on the results and performing extra simulations. However, there are some parts of the major concerns that the authors failed to address fully and there seems to be a few more major points to be addressed.

Major points:

1. In major point 4 in my previous review, I suggested the authors to examine the effect of NMDA receptor kinetics and its synaptic weight change on the 'short-term memory' in order to see whether the current model only works for specific arbitrary parameters that the authors used. However, the authors simply included new simulation results where NMDA receptors were replaced with AMPA receptors in Fig. 5F, to demonstrate that NMDA receptors they used in their study are critical in simulating the phenomenon, hence did not fully address our major point. Moreover, including the AMPA receptor simulation results in Fig 5F is completely unrelated to the other results presented in Fig. 5 thus interferes seriously with the logical flow of the presented data. Instead, they could include the AMPA receptor simulation results (Fig. 5F) as a separate supplementary figure. I still believe that it is important for readers to be presented with simulation data illustrating how the NMDA receptor kinetics and their synaptic weight changes would affect the activity bump directly, to demonstrate that this phenomenon is robust and not specific for certain parameter domain the authors chose. Similar for acetylcholine receptors.
2. It was suggested in previous major point 5 that the role of GABAergic ring neuron should be more clearly justified, however, the authors did address this concern fully either. Instead, they simply rearranged the layout of the Fig. 5. I am not at all convinced by this and would like to ask the authors again to directly demonstrate the effect of GABAergic ring neuron lesion by including the control data of full EB-PB circuit model on the task condition 3 in Fig. 5A. Moreover, in the modeling of full EB-PB circuit model, authors made GABAergic ring neuron alternatively inhibit C-ring and P-ring per each behavior condition. But we could not find any biological evidence for this. Thus, it would be better if the authors firstly introduce the model with only both C-ring and P-ring circuit, and subsequently introduce the model including each types of ring neuron.
3. Authors compared the role of C-ring and P-ring in Fig. 3 and Fig. 4 and found that activity bump of C-ring is more stable and accurate than that of P-ring, and only P-ring is able to track angle in the dark. However, the comparison was not clearly shown in figure. It would be more clear if the authors include the data (Bump center deviation, Bump FWHM) of C-ring on Fig. 4D-E to directly compare the data of C-ring to that of P-ring. Then, the differential role of C-ring and P-ring would become more convincing.
4. In Fig. 4, although P-ring seems less stable and less accurate than C-ring, P-ring sustained the persistent activity bump in condition 1 and was able to track the shifted visual cue in condition 2 as C-ring. Thus, only P-ring might able to track the body orientation during random walk shown in Fig. 5D. Hence, it will be important to clearly show that C-ring and P-ring model is incapable in tracking the body orientation during random walk condition before Figure 5. It is necessary that the direct demonstration why full EB-PB circuit model containing both C-ring and P-ring is required

Minor points:

1. The authors added more than 100 lines to the main text in different parts of the manuscript to address the reviewers comments but in parts, they are not well integrated with the main text and sometimes interrupts the flow of the logic, thus a more careful revision of the manuscript should be done.
2. In line 147, the word ("bump-shifting" function) was firstly introduced. This needs to be clarified in introduction. Moreover, since it is often confused with the term "shift of activity bump for tracking visual cue", the revision of term or the clear introduction of bump-shifting will

be required

3. In line 216-218, the stimulation method for body rotation was firstly introduced. It should be introduced in line 183-187 paragraph.

4. It is difficult to compare/contrast Figures 3D-E and Figures 4D-E, as they are plotted in different figures and also since no statistical tests as been perform. Maybe something could be done to improve the direct comparison.

5. Several typos:

A. Figure 5: EB-FB system \diamond EB-PB system

B. Line 47: the ellipsoid body \diamond EB

C. Line 820: inDrosophila \diamond in Drosophila

Reviewer #3 (Remarks to the Author):

This revision has improved the paper, in particular making some of the modelling decisions and their consequences much clearer.

However I do not feel my original criticisms have been fully addressed.

First, the beginning sentence of the abstract and beginning paragraph of the introduction are still problematic in relating 'goal-directed behaviour' to orientation estimation and 'angular path integration'. Unless a goal is at an effectively infinite distance from the animal, then tracking the animal's orientation changes will not be sufficient for it to maintain an estimate even of the relative orientation of the goal (let alone memory of its 'position'); for nearby goals, only actual path integration (i.e. taking translation also into account) will suffice. In both the evidence from the fly available to date, and in their model, the 'bump' has not been shown to be tracking the position of an external landmark, but only the orientation of the animal relative to the external world, driven by either visual or proprioceptive information about its rotation. Lines 34-45 seem to mix up 'position' and 'orientation' in a random way. I would prefer they dropped all mention of 'goals', 'positions' and 'path integration' and focus instead on 'orientation estimation' as the function of this circuit.

Regarding actual path integration, I agree with the other reviewer that for future work, dealing with translation seems of higher priority than extending to 3D (lines 356-358). The points made from lines 360-391 seem to me much too vague, postulating extension to other CX components or downstream processing to deal with difficult computational problems without providing any sense of how this could be plausible, and I would recommend shortening this substantially.

Their response regarding 10 neurons of each type did not answer my question: are there actually multiples of each type of neuron in the anatomy? Or only one? Or do we not know the answer to this question? If the last is the case, the points they make seem to be implying that their model is *predicting* that there must be multiples (necessary for firing rate, reliability and redundancy). Is that the correct interpretation?

Similarly, I don't feel very convinced by their response that the variable location of the bump can be explained by "a circuit upstream...that flexibly remaps"; that is just a redescription of the phenomena. In fact, ring attractor dynamics can easily be set up that spontaneously produce a bump at an arbitrary location, which can subsequently be shifted in a coherent manner by input; and if it was assumed the visual input is actually providing rotational information that shifts the bump, rather than a direct 'target location' for the bump, this could explain the observed results in the fly. Could they model the system in this way? In the circumstances I would not expect them to actually do so in a revised paper, but the possibility should be mentioned in the discussion.

Response to reviewer comments

We thank the reviewers for their comments. In this second round of revision we have carefully and completely addressed all the points raised by the reviewers. In addition to correcting the typos, we have also added two new figures (Figure 5 and Figure 7). The original Figure 5 became Figure 6 and the original Figure 6 became Figure 8. We also revised several parts of the main text and marked them in red. See below for our point-by-point reply.

Reviewer #1 (Remarks to the Author):

Overall, my points of critique have been adequately addressed in text and figures. I have only minor suggestions to some revised parts of the text.

l.66: "At the single-cell level..."

l.171: "...0.5 EB regions per ..."

l.174: "...as a series of cue-offset events each followed by a cue-onset event at an adjacent position."

l.175/176: "...while it suppresses..."

l.251: "...ring-neuron types..."

l.326: "...by the efference copy or a reafferent signal of the rotation..."

l.338: "...into angular velocity information."

l.346/347: "synaptic plasticity will take place" To me "synaptic plasticity" seems rather a property of a neuron than a process. "plastic changes in synaptic strength"?

l.349: "...bump in the EB..."

l.353: "...to the fruit fly's..."

l.363: "...in the EB..."

l.374: "...state-dependent responses..."

l.377: "...space is represented..."

l.391: "...cellular level."

l.417: "...such a-typical..." or "...non-typical..."

l.425: "...EB play a crucial role..."

l.427: "...that the mutual excitation..."

l.428: Not the markers are expressed; "constituents of the ACh machinery"?

l.432: Neurotransmitter-receptors rather than neurotransmitters determine whether a synapse is excitatory or inhibitory. Please rephrase.

l.531: "3) it does make sense..."

l.558: "...that in locusts..." or "in the locust"

l.562: "...future study identifies..."

l.840: "...innervate the C ring..."

l.883: "...sub-circuit."

Fig.3G right axis labels need more space.

l.1002: "...two ring-neuron classes."

Reply: We are glad that the reviewer is satisfied with our revision. We have addressed all of the minor points raised by the reviewer.

Reviewer #2 (Remarks to the Author):

The authors made a lot of effort to address the major concerns that were raised previously by adding clearer explanations on the results and performing extra simulations. However, there are some parts of the major concerns that the authors failed to address fully and there seems to be a few more major points to be addressed.

Major points:

1. In major point 4 in my previous review, I suggested the authors to examine the effect of NMDA receptor kinetics and its synaptic weight change on the 'short-term memory' in order to see whether the current model only works for specific arbitrary parameters that the authors used. However, the authors simply included new simulation results where NMDA receptors were replaced with AMPA receptors in Fig. 5F, to demonstrate that NMDA receptors they used in their study are critical in simulating the phenomenon, hence did not fully address our major point. Moreover, including the AMPA receptor simulation results in Fig 5F is completely unrelated to the other results presented in Fig. 5 thus interferes seriously with the logical flow of the presented data. Instead, they could include the AMPA receptor simulation results (Fig. 5F) as a separate supplementary figure. I still believe that it is important for readers to be presented with simulation data illustrating how the NMDA receptor kinetics and their synaptic weight changes would affect the activity bump directly, to demonstrate that this phenomenon is robust and not specific for certain parameter domain the authors chose. Similar for acetylcholine receptors.

Reply: To address the reviewer's request, we performed a number of new simulations. NMDA receptors have three unique properties: a large time constant, $[Mg^{2+}]$ block and response saturation (see equations in lines 520 and 522). Therefore, we tested that how changes of each property affect the dynamics of the model circuit. The simulation result, which is included in a new figure (Figure 7) and described in lines 296-331, indicated that the time constant is crucial for maintaining a stable activity bump. An activity bump is difficult to be established if the time constant is smaller than 50 ms. The result is consistent with the current notion of the

role of NMDA in the working-memory and attractor neural circuit. Basically, a large receptor time constant leads to a smooth and continuous synaptic current between input spikes while a short time constant produces an intermittent synaptic current which causes instability of the activity bump.

We also found that the response saturation is not required for maintaining a stable bump but is important for bump shifting during body rotation. Next, we tested the robustness of the circuit by changing the synaptic weight for the NMDA and acetylcholine synapses. The result (also included in Figure 7) indicated that the circuit works reasonably well in a moderate range of NMDA synaptic weight (100%-120% of the original value). Therefore, the model does not rely on a specific and narrowly tuned parameter set. As for the cholinergic synapses, they are modeled as typical ionotropic channels and are not endowed with any unique kinetic property as NMDA are. Therefore, here we tested how the weight of cholinergic synapses affect the model performance. Our result (Figure 7E) showed that the model is insensitive to it. This is simply because the cholinergic synapses were only implemented in the visual and unilateral input to the PB and these synapses do not directly interact with the internal dynamics of the model. Therefore, changes of the Ach weight do not make much impact to the system. On the other hand, the model is robust against changes in the strength of the external input.

2. It was suggested in previous major point 5 that the role of GABAergic ring neuron should be more clearly justified, however, the authors did address this concern fully either. Instead, they simply rearranged the layout of the Fig. 5. I am not at all convinced by this and would like to ask the authors again to directly demonstrate the effect of GABAergic ring neuron lesion by including the control data of full EB-PB circuit model on the task condition 3 in Fig. 5A. Moreover, in the modeling of full EB-PB circuit model, authors made GABAergic ring neuron alternatively inhibit C-ring and P-ring per each behavior condition. But we could not find any biological evidence for this. Thus, it would be better if the authors firstly introduce the model with only both C-ring and P-ring circuit, and subsequently introduce the model including each types of ring neuron.

Reply: We added a new figure (Figure 5, the original Figure 5 became Figure 6) to address the reviewer's comment. In Figures 5B, we demonstrated that without any ring neurons (C ring + P ring only), the circuit simply could not form any activity bump. Adding R_EIP ring neurons (Figure 5C) allows the formation of the bump but it does not shift with body rotation. Adding R_PEI or R_PEN (Figures 5D & E) without R_EIP did not work because no activity bump can be formed. Therefore R_EIP is the most crucial ring neuron type. Next, In Figure 5F, we show that only with all components present (the full model), the circuit can work perfectly. In Figures 5G-I (formerly Fig 5A) we demonstrate the network activity with only two functioning ring neuron types. R_EIP+R_PEN (Figure 5H) provides a condition with bump formation and a

functioning C ring (P ring is suppressed). This condition leads to a static and non-shifting bump. In contrast, R_EIP+R_PEI (Figure 5I) leads to an activity bump that can be shifted by simulated body rotation because of the functioning P ring. However, the bump is much broader than that observed experimentally. In contrast, the full model (Figure 5F) provides a sharper bump that track the body movement more accurately. We also rewrote the related text in lines 247-271. A quantitative comparison between the performance of the full model and P-ring only model is provided in Figure 6E.

3. Authors compared the role of C-ring and P-ring in Fig. 3 and Fig. 4 and found that activity bump of C-ring is more stable and accurate than that of P-ring, and only P-ring is able to track angle in the dark. However, the comparison was not clearly shown in figure. It would be more clear if the authors include the data (Bump center deviation, Bump FWHM) of C-ring on Fig. 4D-E to directly compare the data of C-ring to that of P-ring. Then, the differential role of C-ring and P-ring would become more convincing.

Reply: We thank the reviewer for the suggestion and we have include the data of C-ring on Fig 4D-E to directly compare the two rings.

4. In Fig. 4, although P-ring seems less stable and less accurate than C-ring, P-ring sustained the persistent activity bump in condition 1 and was able to track the shifted visual cue in condition 2 as C-ring. Thus, only P-ring might able to track the body orientation during random walk shown in Fig. 5D. Hence, it will be important to clearly show that C-ring and P-ring model is incapable in tracking the body orientation during random walk condition before Figure 5. It is necessary that the direct demonstration why full EB-PB circuit model containing both C-ring and P-ring is required

Reply: To address the reviewer's comment, we performed new simulations and include the result in Figure 6E (bottom right panel), in which we showed the mean deviation between the bump position and the actual body orientation for the full model and the P-ring only model. The P-ring model produced a much larger deviation than the full model did, indicating a poor performance of the P-ring model in tracking the orientation during random walk. Therefore, the combination of the C and P rings in the full model helps with bringing the deviation and the bump width to the values that are comparable to those observed experimentally. We did not test C ring in such a condition because the activity bump in C-ring simply do not shift at all with the simulated body rotation as indicated in Fig 3F.

Minor points:

1. The authors added more than 100 lines to the main text in different parts of the manuscript

to address the reviewers comments but in parts, they are not well integrated with the main text and sometimes interrupts the flow of the logic, thus a more careful revision of the manuscript should be done.

Reply: We thank the reviewer for the comment. We have carefully revised the manuscript to ensure that the added text is well integrated with the main text. We rewrote the text associated with the Figures 5 and 6 and now we demonstrate the function of each individual ring neuron type before introducing the full model. This makes the flow of the logic more smoothly.

2. In line 147, the word (“bump-shifting” function) was firstly introduced. This needs to be clarified in introduction. Moreover, since it is often confused with the term “shift of activity bump for tracking visual cue”, the revision of term or the clear introduction of bump-shifting will be required

Reply: We now introduce the term “bump-shifting” in introduction (lines 40-43). The terms “bump-shifting” or “shift of activity bump,” describes the movement of the activity bump from one EB location to another. Such shifting can occur due to the movement of a visual cue or due to the rotation of the body in darkness. We used the term “shift/shifting” for both conditions. To avoid confusion, now we use “shift/shifting” specifically for that caused by body rotation in darkness.

3. In line 216-218, the stimulation method for body rotation was firstly introduced. It should be introduced in line 183-187 paragraph.

Reply: We thank the reviewer for bringing up this issue. However, to comprehend the method, the readers first need to know the P-ring architecture, which is introduced after the line 197. Therefore, it may not be a good idea to move a detailed description of the method to lines 183-187. But we also agree with the reviewer that it is odd to use the method first (in lines 183-187) before explaining it in detail (in lines 216-218). To resolve this issue, we moved the detailed explanation of the method together with some description of the P-ring architecture to the method section (lines 619-628), and refer the readers to it when we mention the method in lines 183-187 (now lines 183-184) and lines 216-218 (now 214-218).

4. It is difficult to compare/contrast Figures 3D-E and Figures 4D-E, as they are plotted in different figures and also since no statistical tests as been perform. Maybe something could be done to improve the direct comparison.

Reply: We now directly compare both data in Figures 4D-E. We also performed the statistical analysis and include the result in lines 203-207. The bump deviation and width are significantly larger in the P-ring than in the C-ring.

5. Several typos:

A. Figure 5: EB-FB system □ EB-PB system

B. Line 47: the ellipsoid body □ EB

C. Line 820: inDrosophila □ in Drosophila

Reply: We have corrected all typos. Thanks.

Reviewer #3 (Remarks to the Author):

This revision has improved the paper, in particular making some of the modelling decisions and their consequences much clearer.

However I do not feel my original criticisms have been fully addressed.

1. First, the beginning sentence of the abstract and beginning paragraph of the introduction are still problematic in relating 'goal-directed behaviour' to orientation estimation and 'angular path integration'. Unless a goal is at an effectively infinite distance from the animal, then tracking the animal's orientation changes will not be sufficient for it to maintain an estimate even of the relative orientation of the goal (let alone memory of its 'position'); for nearby goals, only actual path integration (i.e. taking translation also into account) will suffice. In both the evidence from the fly available to date, and in their model, the 'bump' has not been shown to be tracking the position of an external landmark, but only the orientation of the animal relative to the external world, driven by either visual or proprioceptive information about its rotation. Lines 34-45 seem to mix up 'position' and 'orientation' in a random way. I would prefer they dropped all mention of 'goals',

'positions' and 'path integration' and focus instead on 'orientation estimation' as the function of this circuit.

Reply: We thank the reviewer for the comment. We would like to stress that the term “goal-directed behavior” does not necessarily mean “moving toward a destination” but simply means “purposed movement.” However, to avoid the confusing, we dropped all mention of “goals”, “positions” and “integration” from the first paragraph. We rewrote the first couple sentences and now focus on “orientation” as the reviewer suggested.

2. Regarding actual path integration, I agree with the other reviewer that for future work, dealing with translation seems of higher priority than extending to 3D (lines 356-358). The

points made from lines 360-391 seem to me much too vague, postulating extension to other CX components or downstream processing to deal with difficult computational problems without providing any sense of how this could be plausible, and I would recommend shortening this substantially.

*Their response regarding 10 neurons of each type did not answer my question: are there actually multiples of each type of neuron in the anatomy? Or only one? Or do we not know the answer to this question? If the last is the case, the points they make seem to be implying that their model is *predicting* that there must be multiples (necessary for firing rate, reliability and redundancy). Is that the correct interpretation?*

Reply: We have considerably shortened the text in lines 356-391 (now in lines 429-439) and move the translation part forward (now in lines 416-428) to emphasize it. Regarding the question about 10 neurons in each type, we do not know the answer. Yes, it is our model requirement, or prediction in some sense, that each neuron type contains several isomorphic neurons. In fact, some anatomical studies (Young and Armstrong. J. Comp. Neurol. 518, 1500-1524 (2010) for example) indicated multiple isomorphic neurons in some central complex neuron types. We mentioned three reasons to have multiple isomorphic neurons in each type we mentioned in lines 574-586. The first reason is required in order for the model to work and it is the model predictions in some sense. The rest two reasons are purely based on functional consideration. However, if the neurons in the Drosophila brain are endowed with certain slow component in the membrane dynamics, one single neuron per type may be enough. In this case, we only need to update the neuron membrane model and the number of neurons. The main conclusion (functions of C ring and P ring) will not be changed.

3. Similarly, I don't feel very convinced by their response that the variable location of the bump can be explained by "a circuit upstream...that flexibly remaps"; that is just a redescription of the phenomena. In fact, ring attractor dynamics can easily be set up that spontaneously produce a bump at an arbitrary location, which can subsequently be shifted in a coherent manner by input; and if it was assumed the visual input is actually providing rotational information that shifts the bump, rather than a direct 'target location' for the bump, this could explain the observed results in the fly. Could they model the system in this way? In the circumstances I would not expect them to actually do so in a revised paper, but the possibility should be mentioned in the discussion.

Reply: We agree that what the reviewer suggested is one plausible mechanism for the flexible mapping and the model could be implemented that way. We have added it in discussion in lines 403-407.

Reviewers' Comments:

Reviewer #2 (Remarks to the Author):

The authors have adequately addressed the major concerns that I raised in the previous round of the review by conducting more simulations and adding more figures.

However, my concern is that the writing of the main text still needs substantial revision and lack consistency. Also, the newly added sentences and paragraphs still do not integrate well with the text and are written as if they are “stand-alone” paragraphs without stating clearly the purpose as to why certain investigations were carried out. (see point 4). Please inform the readers the purpose of each simulation conditions specifically in each paragraph at the outset otherwise, the results sections are written as a parallel collection of various investigations without being integrated into one big story. I highly recommend thorough revision and also English editing by a native speaker before the manuscript is published.

1. Lines 144-145: “This ... function” → I don't think this sentence is rightly placed. I recommend removing it otherwise the authors should provide more evidence as to why they come to such hypothesis more specifically.
2. Lines 169-170: “The ability of the C ring...speed” & lines 174 – 176: “This process ... second” → In Fig. 3 or in the legend, there is no mention what so ever on the fact that the ability of the C ring to track a moving cue depends on the speed of the cue but somehow, the authors write in their main text in the above sentences as if they have investigated this systematically. Also, the way they express the speed is different in the main text and in the legend. Either include relevant data in Fig. 3 or give reasons specifically why they suddenly investigate the dependence on “speed” without showing the data.
3. Line 211: “Instead, ...” → I recommend this sentence to be rephrased as “Based on our results, we postulated..”
4. Lines 296 – 331 → In these paragraphs, the authors explain the roles of NMDA kinetics in the model in relation to their new simulation result. However, the authors do not clearly write “why” they suddenly decided to investigate the roles of NMDA kinetics. Same with the acetylcholine as well. I know why the authors have included this section in their manuscript but as a reader, I would be puzzled as to why they are doing these investigations since they simply state the results without clearly stating the purpose.
5. In describing their results throughout the text, the authors intermix both “past” and “present” tense. They should choose only one for consistency throughout the text.

Typos:

1. line 36: updates → update
2. line 196: “We then tested” → “We next tested”

Reviewer #3 (Remarks to the Author):

The revised paper has sufficiently answered my previous criticisms. I still feel there are number of aspects of this model (such as number of neurons) which are somewhat arbitrary. The importance of these to the model functioning is crucial, i.e., are they necessary and hence effectively predictions for the animal, or can alternatives be imagined? But I think this is better pursued by publishing and follow-up then repeated cycles through review.

Response to reviewer comments

We thank the reviewers for providing constructive comments. We have address all of them and please see below for our point by point responses.

Reviewer #2 (Remarks to the Author):

The authors have adequately addressed the major concerns that I raised in the previous round of the review by conducting more simulations and adding more figures.

However, my concern is that the writing of the main text still needs substantial revision and lack consistency. Also, the newly added sentences and paragraphs still do not integrate well with the text and are written as if they are “stand-alone” paragraphs without stating clearly the purpose as to why certain investigations were carried out. (see point 4). Please inform the readers the purpose of each simulation conditions specifically in each paragraph at the outset otherwise, the results sections are written as a parallel collection of various investigations without being integrated into one big story. I highly recommend thorough revision and also English editing by a native speaker before the manuscript is published.

Reply: We have addressed all the comments (see below). We also had the manuscript edited by a native speaker.

1. *Lines 144-145: “This ... function” → I don’ t think this sentence is rightly placed. I recommend removing it otherwise the authors should provide more evidence as to why they come to such hypothesis more specifically.*

Reply: This sentence was added to address the comment from one of the reviewers. The reviewer asked us to include some information regarding the putative functions after introducing each ring circuit. I understand that this sentence may disrupts the logic flow of the paragraph. So, I rewrote the sentence and rearranged the order of the last two sentences to make the flow smoother (lines 127-131).

2. *Lines 169-170: ” The ability of the C ring ...speed” & lines 174 – 176: “This process ... second” → In Fig. 3 or in the legend, there is no mention what so ever on the fact that the ability of the C ring to track a moving cue depends on*

the speed of the cue but somehow, the authors write in their main text in the above sentences as if they have investigated this systematically. Also, the way they express the speed is different in the main text and in the legend. Either include relevant data in Fig. 3 or give reasons specifically why they suddenly investigate the dependence on “speed” without showing the data.

Reply: We have included the data in Fig 3C to show how the system barely tracks the moving cue when its speed doubled as described in text.

3. *Line 211: “Instead, ...” → I recommend this sentence to be rephrased as “Based on our results, we postulated..”*

Reply: We thank the reviewer for the suggestion. We have made the change.

4. *Lines 296 – 331 → In these paragraphs, the authors explain the roles of NMDA kinetics in the model in relation to their new simulation result. However, the authors do not clearly write “why” they suddenly decided to investigate the roles of NMDA kinetics. Same with the acetylcholine as well. I know why the authors have included this section in their manuscript but as a reader, I would be puzzled as to why they are doing these investigations since they simply state the results without clearly stating the purpose.*

Reply: We have added and revised several sentences to explain why we need to investigate the dynamics of NMDA and cholinergic synapses. Previous studies suggested that NMDA kinetics is crucial in maintaining activity bump in an attractor networks. Our tests supported this notion. We investigated the effect of the weights of NMDA and cholinergic synapses because we would like to test the robustness of the model circuit.

5. *In describing their results throughout the text, the authors intermix both “past” and “present” tense. They should choose only one for consistency throughout the text.*

Reply: Thanks for the comment. We have carefully addressed this issue.

Typos:

1. *line 36: updates → update*
2. *line 196: “We then tested” → “We next tested”*

Reply: These typos have been corrected. Thanks.

Reviewer #3 (Remarks to the Author):

The revised paper has sufficiently answered my previous criticisms. I still feel there are number of aspects of this model (such as number of neurons) which are somewhat arbitrary. The importance of these to the model functioning is crucial, i.e., are they necessary and hence effectively predictions for the animal, or can alternatives be imagined? But I think this is better pursued by publishing and follow-up then repeated cycles through review.

Reply: We agree with the reviewer. We are currently examining as much data as we can obtain in order to answer these questions. We will address them in the follow-up study.